# GEOMOL: Torsional Geometric Generation of Molecular 3D Conformer Ensembles

**Octavian-Eugen Ganea** [*,‡]          **Lagnajit Pattanaik** [*,†]

**Connor W. Coley** [†]          **Regina Barzilay** [‡]          **Klavs F. Jensen** [†]

**William H. Green** [†]          **Tommi S. Jaakkola** [‡]

## Abstract

Prediction of a molecule's 3D conformer ensemble from the molecular graph holds a key role in areas of cheminformatics and drug discovery. Existing generative models have several drawbacks including lack of modeling important molecular geometry elements (e.g., torsion angles), separate optimization stages prone to error accumulation, and the need for structure fine-tuning based on approximate classical force-fields or computationally expensive methods. We propose GEOMOL — an end-to-end, non-autoregressive, and SE(3)-invariant machine learning approach to generate distributions of low-energy molecular 3D conformers. Leveraging the power of message passing neural networks (MPNNs) to capture local and global graph information, we predict local atomic 3D structures and torsion angles, avoiding unnecessary over-parameterization of the geometric degrees of freedom (e.g., one angle per non-terminal bond). Such local predictions suffice both for both the training loss computation and for the full deterministic conformer assembly (at test time). We devise a non-adversarial optimal transport based loss function to promote diverse conformer generation. GEOMOL predominantly outperforms popular open-source, commercial, or state-of-the-art machine learning (ML) models, while achieving significant speed-ups. We expect such differentiable 3D structure generators to significantly impact molecular modeling and related applications. [4]

## 1 Overview

**Problem & importance.**   We tackle the problem of molecular conformer generation (MCG), i.e., predicting the ensemble of low-energy 3D conformations of a small molecule solely based on the molecular graph (fig. 1). A single conformation is represented by the list of 3D coordinates for each atom in the respective molecule. In this work, we assume that the low-energy states are implicitly defined by the given dataset, i.e., our training data consist of molecular graphs and corresponding sets of energetically favorable 3D conformations. Low-energy structures are the most stable configurations and, thus, expected to be observed most often experimentally.

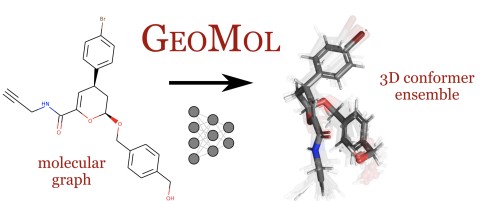

Figure 1: We generate a representative set of low-energy 3D conformers from the input molecular graph. This example molecule has both rigid (rings) and flexible parts. Conformers are shown aligned and juxtaposed.

---

[*]Equal first authorship and contribution. Correspondence to {oct,lagnajit}@mit.edu .

[†]Department of Chemical Engineering, MIT, Cambridge, MA 02139

[‡]Computer Science and Artificial Intelligence Laboratory, MIT, Cambridge, MA 02139

[4]Code is available at https://github.com/PattanaikL/GeoMol.

35th Conference on Neural Information Processing Systems (NeurIPS 2021).

Dealing with molecules in their natural 3D structure is of great importance in areas such as cheminformatics or computational drug discovery because conformations determine biological, chemical, and physical properties [Guimaraes et al., 2012, Schütt et al., 2018, Klicpera et al., 2019, Axelrod and Gomez-Bombarelli, 2020b, Schütt et al., 2021, Liu et al., 2021] such as charge distribution, potential energy, docking poses [McGann, 2011], shape similarity [Kumar and Zhang, 2018], pharmacophore searching [Schwab, 2010], or descriptors for 3/4D QSAR [Verma et al., 2010]. For instance, in drug design it is crucial to understand how a molecule binds to a specific target protein; this process heavily depends on the 3D structures of the two components, both in terms of geometric (shape matching) and chemical (hydrophobic/hydrophilic) interactions [Gainza et al., 2020, Sverrisson et al., 2020].

**Motivation & challenges of existing methods.** The main challenge in MCG comes from the enormous size of the 3D structure space consisting of bond lengths, bond angles, and torsion angles. It is known that the molecular graph imposes specific constraints on possible 3D conformations, e.g., bond length ranges depend on the respective bond types, while tetrahedral centers dictate local spatial arrangement. However, the space of possible conformations grows exponentially with the graph size and number of rotatable bonds, thus hindering exhaustive brute force exploration even for relatively small molecules. Additionally, the number of plausibly-stable low-energy states is unknown a priori and can vary between one and several thousand conformations for a single molecule [Chan et al., 2021]. Nevertheless, various facets of the curse of dimensionality have been favorably tackled by ML models in different contexts, and our goal is to build on the recent ML efforts for MCG [Mansimov et al., 2019, Simm and Hernandez-Lobato, 2020, Lemm et al., 2021, Xu et al., 2021].

Molecular conformations can be determined experimentally, but existing techniques are very expensive. As a consequence, predictive computational models have been developed over the past few decades, traditionally being categorized as either *stochastic* or *systematic* (rule-based) methods [Hawkins, 2017]. Stochastic approaches have traditionally been based on molecular dynamics (MD) or Markov chain Monte Carlo (MCMC) techniques, potentially combined with genetic algorithms (GAs). They can do extensive explorations of the energy landscape and accurately sample equilibrium structures, but quickly become prohibitively slow for larger molecules [Shim and MacKerell Jr, 2011, Ballard et al., 2015, De Vivo et al., 2016, Hawkins, 2017], e.g., they require several CPU minutes for a single drug-like molecule. Moreover, stochastic methods have difficulties sampling diverse and representative conformers, prioritizing quantity over quality. On the other hand, rule-based systematic methods achieve state-of-the-art in commercial software [Friedrich et al., 2017] with OMEGA [Hawkins et al., 2010, Hawkins and Nicholls, 2012] being a popular example. They usually process a single drug-like molecule under a second. They address the aforementioned challenges of stochastic methods by relying on carefully curated torsion templates (torsion rules), rule-based generators, and knowledge bases of rigid 3D fragments, which are assembled together and combined with subsequent stability score ranking. However, torsion angles are mostly varied independently (based on their fragments), without explicitly capturing their global interactions, which results in difficulties for larger and more flexible molecules. Furthermore, the curated fragments and rules are inadequate for more challenging inputs (e.g., transition states or open-shell molecules).

Both types of methods can be combined with Distance Geometry (DG) techniques to generate the initial 3D conformation. First, the 3D atom distance matrix is generated based on a set of distance constraints or from a specialized model. Subsequently, the corresponding 3D atom coordinates are learned to approximately match these predicted distances [Havel et al., 1983b,a, Crippen et al., 1988, Havel, 1998, Lagorce et al., 2009, Riniker and Landrum, 2015]. Indeed, modern stochastic algorithms are entirely based on DG methods [Riniker and Landrum, 2015]. The inductive bias of rotational and translational invariance is guaranteed for DG, thus being appealing for ML models [Simm and Hernandez-Lobato, 2020, Xu et al., 2021, Pattanaik et al., 2020b]. However, several drawbacks weaken this important direction: i) the distance matrix is overparameterized compared to the actual number of degrees of freedom, ii) it is difficult to enforce 3D Euclidean distance constraints as well as geometric graph constraints (e.g., on torsion angles or rings [Riniker and Landrum, 2015]); iii) important aspects of molecular geometry are not explicitly modeled, e.g., torsion angles of rotatable bonds or tetrahedral centers; iv) expensive force-field energy fine-tuning of the generated conformers is vital for a reasonable quality [Xu et al., 2021, Simm and Hernandez-Lobato, 2020]; iv) the resulting multi-stage pipeline is prone to error accumulation as opposed to an end-to-end model.

Previous methods often rely on a force field (FF) energy function minimization to fine-tune the conformers. These are hand-designed energy models which use parameters estimated from experi-

ment and/or computed from quantum mechanics (e.g., Universal Force Field [Rappé et al., 1992], Merck Molecular Force Field [Halgren, 1996]). However, FFs are crude approximations of the true molecular potential energy surface [Kanal et al., 2018], limited in the interactions they can capture in biomolecules due to their strong assumptions [Barman et al., 2015]. In addition, FF energy optimization is relatively slow and increases error accumulation in a multi-pipeline method.

**Relation to protein folding.** There has been impressive recent progress on modeling protein folding dynamics [Ingraham et al., 2018, AlQuraishi, 2019, Noé et al., 2019, Senior et al., 2020], where crystallized 3D structures are predicted solely from the amino-acid sequence using ML methods. However, molecules pose unique challenges, being highly branched graphs containing cycles, different types of bonds, and chirality information. This makes protein folding approaches not readily transferable to general molecular data.

**Our key contributions & model in a nutshell.** In this work, we investigate the question:

*Can we design a fast and generalizable deep learning model to predict high-quality, representative, and diverse 3D conformational ensembles from input molecular graphs?*

To tackle this question, we propose GEOMOL (shown in fig. 2), exhibiting the following merits:

- It is end-to-end trainable, non-autoregressive, and does not rely on DG techniques (thus avoiding aforementioned drawbacks). More precisely, it outputs a minimal set of geometric quantities (i.e., angles and distances) sufficient for full deterministic reconstruction of the 3D conformer.

- It models conformers in an SE(3)-invariant (translation/rotation) manner by design. This desirable inductive bias was previously either achieved using multi-step DG methods [Simm and Hernandez-Lobato, 2020] or not captured at all [Mansimov et al., 2019].

- It explicitly models and predicts essential molecular geometry elements: torsion angles and local 3D structures (bond distances and bond angles adjacent to each atom). Together with the input molecular graph, these are used for k-hop distance computation at train time and full deterministic conformation assembly at test time. Crucially, we do not over-parameterize these predictions, i.e., a single torsion angle is computed per each non-terminal bond, irrespective of the number and permutation of the neighboring atoms at each end-point of the respective bond.

- The above geometric elements (torsion angles, local structures) are SE(3)-invariant (by definition or usage) and we jointly predict them using MPNNs [Gilmer et al., 2017] and self-attention networks. Thus, unlike [Mansimov et al., 2019], we are not affected by MPNNs' pitfalls that obstruct direct predictions of 3D atom coordinates from node embeddings, e.g., symmetric or locally isomorphic nodes would always have identical MPNN embeddings [Xu et al., 2019, Garg et al., 2020] and, as a consequence, would be inappropriately assigned identical 3D coordinates.

- To promote diverse conformer ensembles with good coverage, we devise a tailored generative loss that does not use slow or difficult-to-optimize adversarial training techniques. Using optimal transport, GEOMOL finds the best matching between generated and ground truth conformers based on their pairwise log-likelihood loss, requiring only minimization.

- It explicitly and deterministically distinguishes reflected structures (enantiomers) by solving tetrahedral stereocenters using oriented volumes and local chiral descriptors, bypassing the need for iterative optimization usually done in DG approaches.

- Empirically, we conduct experiments on two benchmarks: GEOM-QM9 (smaller molecules relevant to gas-phase chemistry) and GEOM-DRUGS (drug-like molecules) [Axelrod and Gomez-Bombarelli, 2020a]. Our method often outperforms previous ML and two popular open-source or commercial methods in different metrics. Moreover, we show competitive quality even without the frequently-used computationally-demanding fine-tuning FF strategies.

- GEOMOL processes drug-like molecules in seconds or less, being orders of magnitude faster than popular baselines (e.g., ETKDG/RDKit[Riniker and Landrum, 2015]), without sacrificing quality.

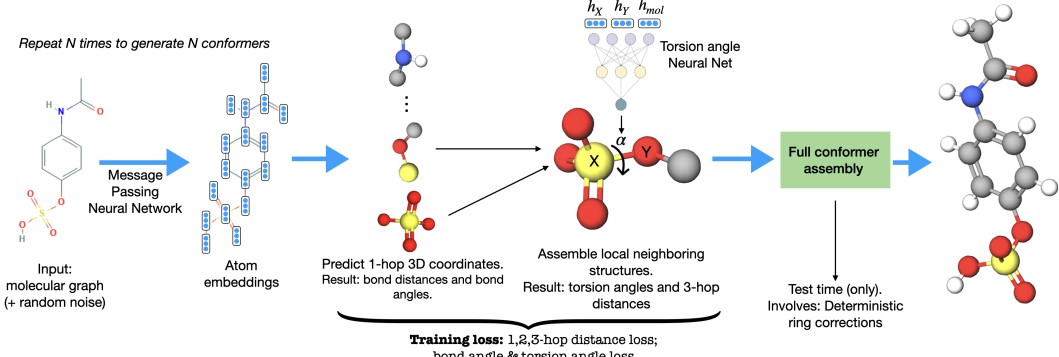

Figure 2: Overview of the GEOMOL model, which is SE(3)-invariant by design. Given a molecular graph, we first compute MPNNs atom embeddings. Next, we predict the local 3D structures (LS) of each non-terminal atom in a permutation invariant way, explicitly solving chirality. Third, for each bond connecting non-terminal vertices, we assemble the two LS by predicting a single torsion angle, avoiding overparameterization. Finally, the full conformer is assembled (only) at test time.

## 2  Method

**Problem setup & notations.**    Our input is any molecular graph $G = (V, E)$ with node and edge features, $\mathbf{x}_v \in \mathbb{R}^f, \forall v \in V$ and $\mathbf{e}_{u,v} \in \mathbb{R}^{f'}, \forall (u,v) \in E$ representing atom types, formal charges, bond types, etc. For each molecular graph G, we have a variable-size set of low-energy ground truth 3D conformers $\{\mathcal{C}_l^*\}_l$ that we predict with a model $\{\mathcal{C}_k\}_k \stackrel{\text{def}}{=} \zeta(G)$. A conformer is a map $\mathcal{C} : V \to \mathbb{R}^3$ from graph nodes to 3D coordinates, but a simplified notation is $\mathbf{c}_v \in \mathbb{R}^3$ for $v \in V$. We use additional notations: $d(X, Y) = \|\mathbf{c}_X - \mathbf{c}_Y\|$ is the 3D distance between X and Y; $\angle XYZ$ is the counter-clockwise (CCW) angle $\angle \mathbf{c}_X \mathbf{c}_Y \mathbf{c}_Z$; $\angle(XYZ, XYT)$ is the CCW dihedral angle of the 2D planes $\mathbf{c}_X \mathbf{c}_Y \mathbf{c}_Z$ and $\mathbf{c}_X \mathbf{c}_Y \mathbf{c}_T$ (formula is in appendix C). We use the corresponding $\mathbf{c}_v^*, d^*(X, Y), \angle^* XYZ, \angle^*(XYZ, XYT)$ when manipulating a ground truth conformer.

Any conformer is defined up to a SE(3) transformation, i.e., any translation or rotation applied to the set $\{\mathbf{c}_v\}_{v \in V}$. A classic conformer distance function that satisfies this constraint is root-mean-square deviation of atomic positions (RMSD), computed by the Kabsch alignment algorithm [Kabsch, 1976].

### 2.1  GEOMOL high-level overview

Our approach, shown in fig. 2, comprises three steps. First, we predict the local 3D structure of each non-terminal atom, which we deem *local structure* (LS), by combining self-attention layers and MPNNs with deterministic corrections for tetrahedral centers. Bond distances and bond angles are computed from the predicted LS. Next, we assemble all neighboring pairs of LSs by predicting the torsion angles and aligning them. Importantly, since LSs are fixed, it suffices to only predict a single value for the dihedral angle of each bond. Towards this goal, we develop a canonical representation of torsion angles via a local coordinate system defined SE(3)-equivariantly w.r.t. the full structure, which allows us to predict exactly the number of degrees of freedom. Finally, at test time, we assemble all predicted pairs of neighboring LSs to construct the full conformer, applying deterministic ring corrections. In order to generate diverse conformers, we append random Gaussian noise vectors to each initial node feature vector and use an optimal transport-based loss function for training.

### 2.2  Message passing neural networks (MPNNs)

Given an input graph G, an MPNN [Gilmer et al., 2017, Battaglia et al., 2018, Yang et al., 2019] computes node embeddings $\mathbf{h}_v \in \mathbb{R}^d, \forall v \in V$ using $T$ layers of iterative message passing:

$$\mathbf{h}_u^{(t+1)} = \psi \left( \mathbf{h}_u^{(t)}, \sum_{v \in \mathcal{N}_u} \phi(\mathbf{h}_v^{(t)}, \mathbf{h}_u^{(t)}, \mathbf{e}_{u,v}) \right), \quad \text{where } \mathbf{h}_v^{(0)} \stackrel{\text{def}}{=} concat[\mathbf{x}_v, \mathbf{z}_v], \ \mathbf{z}_v \sim \mathcal{N}(\mathbf{0}, s\mathbf{I}_d)$$

(1)

for each $t \in [0 \mathinner{..} T-1]$, where $\mathcal{N}_u = \{v \in V | (u,v) \in E\}$, while $\psi$ and $\phi$ are generic functions, e.g., implemented using multilayer perceptrons (MLP) or attention [Veličković et al., 2017]. Final node embeddings are obtained by the embedding of the last layer: $\mathbf{h}_v \stackrel{\text{def}}{=} \mathbf{h}_v^{(T)}, \forall v \in V$. Finally, we

also compute a molecular embedding: $\mathbf{h}_{mol} \overset{\text{def}}{=} MLP(\sum_{v \in V} \mathbf{h}_v)$. We leave comparison with other MPNN variants for future work, e.g., Kipf and Welling [2017], Veličković et al. [2017], Hamilton et al. [2017], Xu et al. [2019].

## 2.3 Local structure (1-hop) prediction model

Following notations in fig. 3, for each non-terminal graph vertex $X \in V$ having $n$ graph neighbors $\mathcal{N}_X = \{T_i\}_{i \in [1..n]}$, we predict its *local 3D structure* (LS), i.e., the relative 3D positions of all $T_i$, when X is centered in the origin. The generic model is a function $f(\mathbf{h}_{T_1}, \ldots, \mathbf{h}_{T_n}; \mathbf{h}_X) = (\mathbf{p}_1, \ldots, \mathbf{p}_n) \in \mathbb{R}^{3 \times n}$ that, additionally, should satisfy permutation equivariance w.r.t. $T_i$'s, namely, the 3D position of each neighbor $T_i$ should not change regardless of the ordering of the $X$'s neighbors:

$$f(\mathbf{h}_{T_{\pi(1)}}, \ldots, \mathbf{h}_{T_{\pi(n)}}; \mathbf{h}_X) = (\mathbf{p}_{\pi(1)}, \ldots, \mathbf{p}_{\pi(n)}), \forall \pi \in S_n \qquad (2)$$

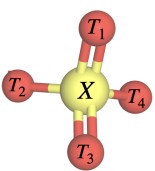

Our choice is the encoder part of a transformer [Vaswani et al., 2017], without any positional encoding, thus satisfying permutation equivariance. This model takes as input the set $\{concat[\mathbf{h}_{T_i}, \mathbf{h}_X]; i \in [1..n]\}$ *in any order* and synchronously updates the $n$ embeddings based on several transformer layers. The final layer projects the embeddings to 3 dimensions, resulting in a list $(\mathbf{p}_1, \ldots, \mathbf{p}_n) \in \mathbb{R}^{3 \times n}$ having the exact same node order as the input list.

Figure 3: For each non-terminal atom X, we predict the relative 3D position of each of its graph neighbors, $\{T_i\}_{i \in [1..n]}$, in a permutation equivariant manner.

**Enforcing local consistency.** We desire the LS model $f()$ to be *distance-consistent*, i.e., any bond distance $d(X, Y)$ is the same, no matter if it is computed from the LS of node X or of node Y. To achieve this, we use the above transformer just to compute bond directions (which will be aligned using a separate approach described in section 2.4), while we obtain the bond distances with a separate symmetric model. Concretely, let the above transformer $f()$ predict $(\mathbf{p}_1, \ldots, \mathbf{p}_n) \in \mathbb{R}^{3 \times n}$, while the final local 3D coordinates are $\mathbf{p}'_i \overset{\text{def}}{=} \frac{\mathbf{p}_i}{\|\mathbf{p}_i\|} d_{GNN}(\mathbf{h}_X, \mathbf{h}_{T_i}), \forall i$, where each bond distance is predicted with a symmetric model $d_{GNN}(\mathbf{h}_X, \mathbf{h}_Y) \overset{\text{def}}{=} \text{softplus}(\psi(\mathbf{h}_X, \mathbf{h}_Y) + \psi(\mathbf{h}_Y, \mathbf{h}_X)), \forall (X, Y) \in E$, with the same shared $\psi$ (e.g., an MLP). For notation simplicity, we will just use $\mathbf{p}_i$ instead of $\mathbf{p}'_i$.

**Regarding SE(3) invariance.** The above model is not SE(3)-invariant *per se*, but it is used as such. Namely, on one hand we compute SE(3)-invariant quantities: 1-hop distances $d(T_i, X)$, 2-hop distances $d(T_i, T_j)$, and bending angles $\angle T_i X T_j$. These will be compared to their ground-truth counterparts in the final loss, see section 2.5.

On the other hand, the LS of adjacent graph nodes are assembled together for computing torsion angles or for building the full conformer at test time. This process is explicitly defined to be SE(3)-invariant as described in section 2.4.

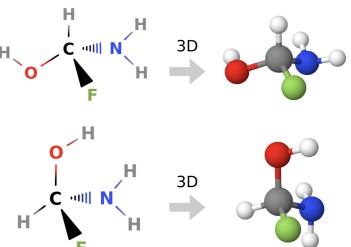

**Tetrahedral chiral corrections.** When embedding the local neighborhood of a node in 3D space, one has to carefully account for tetrahedral stereocenters (fig. 4). Tetrahedral chirality is a common form of stereochemistry which restricts the 3D location of neighboring substituents of a central atom with four distinct neighbors; molecules which differ by a single tetrahedral stereocenter, i.e., enantiomers, are mirror images of each other. Chirality heavily impacts some properties of small molecules–e.g., bioactivity. Existing MPNNs using only the molecular graph cannot distinguish chiral centers (fig. 4), but solutions exist [Pattanaik et al., 2020a]. Mathematically, enantiomers can be differentiated based on the oriented volume around the tetrahedral center. That is, given the ordered set of neighbor 3D coordinates around the center, namely $\mathbf{p}_1, \mathbf{p}_2, \mathbf{p}_3, \mathbf{p}_4 \in \mathbb{R}^3$, the

Figure 4: Chirality: even if the two shown graphs are isomorphic, they have distinct 3D structures that can be distinguished by the order of the carbon center's neighbors.

sign of the volume of the tetrahedron formed by the neighbors is

$$OV(\mathbf{p}_1, \mathbf{p}_2, \mathbf{p}_3, \mathbf{p}_4) \overset{\text{def}}{=} sign \left( \begin{vmatrix} 1 & 1 & 1 & 1 \\ x_1 & x_2 & x_3 & x_4 \\ y_1 & y_2 & y_3 & y_4 \\ z_1 & z_2 & z_3 & z_4 \end{vmatrix} \right)$$

Enantiomeric structures always have opposite signs for the oriented volume [Crippen et al., 1988]. Since we generate local 3D structures directly, we can also use local 3D chiral descriptors to ensure the correct generation of tetrahedral stereocenters. RDKit internally keeps track of these local chiral labels, denoted by CW/CCW labels (detailed in e.g., Pattanaik et al. [2020a]). Importantly, each local chiral label corresponds to a certain oriented volume (CW = +1 and CCW = -1). Thus, when generating an LS for a tetrahedral center, we calculate the oriented volume and check against the internal RDKit label. If it results in the incorrect oriented volume (i.e., the incorrect stereocenter was generated), we simply reflect the structure by flipping against the z-axis. This ensures that all *tetrahedral stereocenters centers are generated exactly*, and no iterative optimization is necessary as with traditional DG-based generators.

## 2.4 Torsion angle representation and local structure (LS) assembly

Once the LS of each atom/vertex is predicted, we assemble them in pairs corresponding to each non-terminal bond in the molecular graph. We describe this process for a bond connecting atoms X and Y, each having additional graph neighbors $\{T_i\}_{i \in [1..n]}$ and, resp., $\{Z_j\}_{j \in [1..m]}$. See fig. 5.

**Torsion angle over-parameterization.** We first note that, for any assembled bond XY (fig. 5, right), and $\forall i, k \in [1..n], \forall j, l \in [1..m]$, the dihedral angles $\angle(XYT_i, XYT_k)$ and $\angle(XYZ_l, XYZ_j)$ are fully determined by the LS of nodes X and Y, respectively, so they do not depend on the torsion angle of bond XY. Next, observe that there is exactly one torsion angle for any bond XY, given unique indexing of the neighbors. This happens because of the following constraint:

$$\angle(XYT_i, XYZ_j) = [\angle(XYT_k, XYZ_l) + \angle(XYT_i, XYT_k) + \angle(XYZ_l, XYZ_j)](\text{mod } 2\pi) \quad (3)$$

Thus, in order to avoid unnecessary over-parameterization, we predict a single torsion angle $\alpha$ per each bond $XY$ connecting non-terminal atoms.

**Torsion angle formulation.** However, it is still unclear at this point how to define this unique angle in a canonical way that is: i) permutation invariant w.r.t. the nodes in the set $\{T_i\}_{i \in [1..n]}$ and, respectively, in the set $\{Z_j\}_{j \in [1..m]}$, ii) SE(3)-invariant w.r.t. the full 3D conformer, and iii) agrees with eq. (3).

Let $\Delta_{ij} \overset{\text{def}}{=} \angle(XYT_i, XYZ_j)$ and $\mathbf{s}_{ij} \overset{\text{def}}{=} \begin{bmatrix} \cos(\Delta_{ij}) \\ \sin(\Delta_{ij}) \end{bmatrix}$. Let $c_{ij} \in \mathbb{R}$ be real coefficients such that $\mathbf{s} \overset{\text{def}}{=} \sum_{i,j} c_{ij}\mathbf{s}_{ij} \in \mathbb{R}^2$ is not the null vector. Then, we define the torsion angle as[5]: $\alpha \overset{\text{def}}{=} atan2(\frac{\mathbf{s}}{\|\mathbf{s}\|})$. It is easy to see that this formulation satisfies both invariances claimed above. We further state (and prove in appendix A) that our proposed formulation gives a torsion angle uniquely determined by all local angles $\angle(YXT_i, YXT_k)$, $\angle(YXZ_j, YXZ_l)$ and by the true underlying torsion angle:

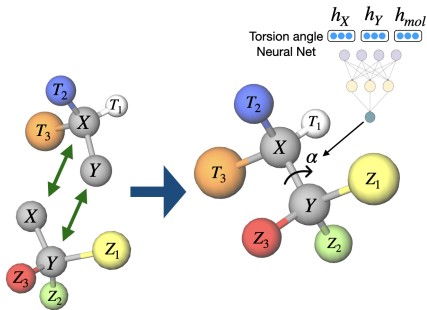

Figure 5: Assembly of the local structures of bonded atoms X and Y based on the predicted torsion angle.

**Proposition 1.** *Given 3D coordinates of nodes $X, Y, T_i, Z_j$ and fixed weights $c_{ij} \in \mathbb{R}$ such that $\sum_{i,j} c_{ij}\mathbf{s}_{ij} \in \mathbb{R}^2$ is not the null vector, then $\alpha \overset{\text{def}}{=} atan2(\frac{\mathbf{s}}{\|\mathbf{s}\|})$ is unique, i.e., if we change the torsion angle of bond XY, then $\alpha$ will change. Formally, if we rotate the set of bonds $\{XT_i\}_i$ jointly around the line XY with the same angle $\gamma$, then $\alpha$ will be exactly shifted with $\gamma$.*

---

[5]We define atan2 slightly different than standard: $atan2(r\cos(\alpha), r\sin(\alpha)) \overset{\text{def}}{=} \alpha, \forall \alpha \in [0, 2\pi), r \in \mathbb{R}_+^*$.

**How to set $c_{ij}$? Breaking symmetries.**     A simple solution is to choose $c_{ij} = 1, \forall i, j$. However, in some important cases, local symmetries may result in $\mathbf{s} = 0$. For example, this happens if, for some $j$, we have $\Delta_{ij} = \frac{2i\pi}{n} + ct., \forall i \in [1..n]$. One solution is to use different $c_{ij}$ to differentiate between the different subgraphs rooted at different $T_i$ (and similarly for $Z_j$). This is reminiscent of traditional group priorities used for distinguishing E/Z isomers. We devise a flexible solution to distinguish these subgraphs: a differentiable real valued function computed from the MPNN node embeddings as $c_{ij} = MLP(\mathbf{h}_{T_i} + \mathbf{h}_{Z_j}) \in \mathbb{R}$, with MLP being a neural network shared across all bonds and molecules. Note that we constrain $c_{ij} = c_{ji}$, thus guaranteeing that the same $\alpha$ is obtained if we swap X and Y (and their neighbors, respectively).

**Final LS assembly for a single bond.**     We now describe the assembly process depicted in fig. 5. We first predict the LS of node X as in section 2.3, obtaining several 3D coordinates: $\mathbf{p}_X = \mathbf{0}, \mathbf{p}_Y, \mathbf{p}_{T_i} \in \mathbb{R}^3, \forall i \in [1..n]$, as well as the LS of node Y: $\mathbf{q}_Y = \mathbf{0}, \mathbf{q}_X, \mathbf{q}_{Z_j} \in \mathbb{R}^3, \forall j \in [1..m]$. By design, we have that $\|\mathbf{q}_X\| = \|\mathbf{p}_Y\|$. These two sets are currently not aligned. To achieve this, we first rotate the LS of X such that $\mathbf{p}_Y$ becomes $[\|\mathbf{p}_Y\| \quad 0 \quad 0]^\top$, while $\mathbf{p}_X$ remains $\mathbf{0}$. Next, we rotate and translate the LS of Y such that $\mathbf{q}_Y$ becomes $\mathbf{p}_Y$ and $\mathbf{q}_X$ becomes $\mathbf{p}_X = \mathbf{0}$. These two rotations have one degree of freedom each, which we set randomly. Exact formulas are in appendix B. Thus, the bond XY is now matched, but the torsional rotation is still arbitrary/random. The remaining step is to rotate the LS of X with an angle $\gamma$ such that all dihedrals $\angle(XYT_i, XYZ_j)$ match their true counterparts. This is done by applying to all vectors $\mathbf{p}_{T_i}$ the same rotation of type: $\mathbf{H}_\gamma := \begin{bmatrix} 1 & 0 & 0 \\ 0 & \cos(\gamma) & -\sin(\gamma) \\ 0 & \sin(\gamma) & \cos(\gamma) \end{bmatrix}$.

*How to compute $\gamma$?* The current dihedrals $\Delta_{ij}^{cur} \overset{\text{def}}{=} \angle^{cur}(XYT_i, XYZ_j)$ depend on the random torsional rotations from the initial assembly step of LS of X and of Y. After applying the $\mathbf{H}_\gamma$ rotation, we obtain the new dihedral angles: $[\Delta_{ij}^{cur} - \gamma] \bmod 2\pi$ that should match the ground truth dihedral angles $\Delta_{ij}^* \overset{\text{def}}{=} \angle^*(XYT_i, XYZ_j)$. This is equivalently written as $\mathbf{s}_{ij}^* = \mathbf{A}_{ij}^{cur}\mathbf{s}_\gamma$, where $\mathbf{s}_\gamma \overset{\text{def}}{=} \begin{bmatrix} \cos(\gamma) \\ \sin(\gamma) \end{bmatrix}$ and $\mathbf{A}_{ij}^{cur} \overset{\text{def}}{=} \begin{bmatrix} \cos(\Delta_{ij}^{cur}) & \sin(\Delta_{ij}^{cur}) \\ \sin(\Delta_{ij}^{cur}) & -\cos(\Delta_{ij}^{cur}) \end{bmatrix}$. Let $\mathbf{s}^* \overset{\text{def}}{=} \sum_{i,j} c_{ij}\mathbf{s}_{ij}^*$ and $\mathbf{A}^{cur} \overset{\text{def}}{=} \sum_{i,j} c_{ij}\mathbf{A}_{ij}^{cur}$. The necessary condition for $\gamma$ becomes $\mathbf{s}_\gamma = (\mathbf{A}^{cur})^\top \mathbf{s}^*$, which is also sufficient due to proposition 1. This implies it is enough to predict *only* the normalized $\frac{\mathbf{s}^*}{\|\mathbf{s}^*\|}$ and, in practice, we do that by predicting $\mathbf{s}_\alpha \overset{\text{def}}{=} \begin{bmatrix} \cos(\alpha) \\ \sin(\alpha) \end{bmatrix}$ using a function commutative in X and Y (i.e., swapping X and Y does not change $\alpha$):

$$\alpha = [\phi(\mathbf{h}_X, \mathbf{h}_Y, \mathbf{h}_{mol}) + \phi(\mathbf{h}_Y, \mathbf{h}_X, \mathbf{h}_{mol})]\bmod 2\pi \tag{4}$$

where $\phi$ is a neural network (e.g., MLP). Finally, $\mathbf{s}_\gamma = \begin{bmatrix} \cos(\gamma) \\ \sin(\gamma) \end{bmatrix} = \frac{1}{\|(\mathbf{A}^{cur})^\top\mathbf{s}_\alpha\|} (\mathbf{A}^{cur})^\top \mathbf{s}_\alpha$.

## 2.5 An optimal transport (OT) loss function for diverse conformer generation

**Loss per single conformer.**     Assume first that we predict a single conformer $\mathcal{C}$. Based on all LS and torsion angle predictions, we deterministically compute all 1/2/3-hop distances and bond/torsion angles. If the corresponding ground truth conformer $\mathcal{C}^*$ is known, we feed those quantities into a negative log-likelihood loss, denote by $\mathcal{L}(\mathcal{C}, \mathcal{C}^*)$ and detailed in appendix D. Similar to Senior et al. [2020], we fit distances using normal distributions and angles using von Mises distributions. This is a much faster approach compared to habitual RMSD losses that compare full conformers.

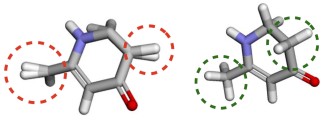

Figure 6: Before (left) and after (right) introducing a matching loss to distinguish symmetric graph nodes. Hydrogen predictions in both groups are visibly improved.

**Node symmetries.**     Our current formulation has difficulties distinguishing pairs of symmetric graph nodes that are less than 3 hops away, e.g., hydrogen groups. We address this using a tailored matching loss detailed in appendix D and exemplified in fig. 6.

Table 1: Results on the **GEOM-DRUGS** dataset. All models are without FF fine-tuning. "R" and "P" denote Recall and Precision. Note: OMEGA is an established commercial (C) software.

| Models | COV - R (%) ↑ | | AMR - R (Å) ↓ | | COV - P (%) ↑ | | AMR - P (Å) ↓ | |
|---|---|---|---|---|---|---|---|---|
| | Mean | Median | Mean | Median | Mean | Median | Mean | Median |
| GraphDG *(ML)* | 10.37 | 0.00 | 1.950 | 1.933 | 3.98 | 0.00 | 2.420 | 2.420 |
| CGCF *(ML)* | 54.35 | 56.74 | 1.248 | 1.224 | 24.48 | 15.00 | 1.837 | 1.829 |
| RDKit/ETKDG | 68.78 | 76.04 | 1.042 | 0.982 | 71.06 | 88.24 | 1.036 | 0.943 |
| OMEGA *(C)* | 81.64 | 97.25 | 0.851 | **0.771** | 77.18 | **96.15** | 0.951 | **0.854** |
| GEOMOL ($s = 9.5$) | **86.07** | **98.06** | **0.846** | 0.820 | 71.78 | 83.77 | 1.039 | 0.982 |
| GEOMOL ($s = 5$) | 82.43 | 95.10 | 0.862 | 0.837 | **78.52** | 94.40 | **0.933** | **0.856** |

Table 2: Results on the **GEOM-QM9** dataset. See caption of table 1.

| Models | COV - R (%) ↑ | | AMR - R (Å) ↓ | | COV - P (%) ↑ | | AMR - P (Å) ↓ | |
|---|---|---|---|---|---|---|---|---|
| | Mean | Median | Mean | Median | Mean | Median | Mean | Median |
| GraphDG *(ML)* | 74.66 | 100.00 | 0.373 | 0.337 | 63.03 | 77.60 | 0.450 | 0.404 |
| CGCF *(ML)* | 69.47 | 96.15 | 0.425 | 0.374 | 38.20 | 33.33 | 0.711 | 0.695 |
| RDKit/ETKDG | 85.13 | **100.00** | 0.235 | 0.199 | **86.80** | **100.00** | 0.232 | 0.205 |
| OMEGA *(C)* | 85.51 | **100.00** | **0.177** | **0.126** | 82.86 | **100.00** | **0.224** | **0.186** |
| GEOMOL ($s = 5$) | **91.52** | **100.00** | 0.225 | 0.193 | **86.71** | **100.00** | 0.270 | 0.241 |

**Total OT loss per ensemble of conformers.** In practice, our model generates a set of conformers $\{\mathcal{C}_k\}_{k \in [1..K]}$ that needs to match a variable sized set of low-energy ground truth conformers, $\{\mathcal{C}_l^*\}_{l \in [1..L]}$. However, we do not know a priori the number $L$ of true conformers or the matching between generated and true conformers. We also wish to avoid expensive and problematic adversarial training. Our solution is an OT-based, minimization-only, loss function:

$$\mathcal{L}^{ensemble} \overset{\text{def}}{=} EMD_{\mathcal{L}(\cdot,\cdot)}(\{\mathcal{C}_k\}_k, \{\mathcal{C}_l^*\}_l) = \min_{\mathbf{T} \in \mathcal{Q}_{K,L}} \sum_{k,l} T_{kl} \mathcal{L}(\mathcal{C}_k, \mathcal{C}_l^*)$$

where EMD is the Earth Mover Distance, $\mathbf{T}$ is the ***transport plan*** satisfying $\mathcal{Q}_{K,L} \overset{\text{def}}{=} \{\mathbf{T} \in \mathbb{R}_+^{K \times L} : \mathbf{T} \mathbf{1}_L = \frac{1}{K} \mathbf{1}_K, \mathbf{T}^T \mathbf{1}_K = \frac{1}{L} \mathbf{1}_L\}$. The minimization w.r.t. $\mathbf{T}$ is computed quickly using Earth Mover Distance and the POT library [Flamary and Courty, 2017].

### 2.6 Full conformer assembly at test time

Knowing all true LSs and torsion angles is, in theory, enough for a deterministic unique SE(3)-invariant reconstruction of the full conformer. However, in practice, these predictions might have small errors that accumulate, e.g., in rings. To mitigate this issue, we deterministically build the full conformer (only at test time) by first predicting a smoothed structure of (fused) rings separately, and then assembling the full conformer following any graph traversal order (any order gives the same conformer, so this procedure does not break the non-autoregressive behavior). We detail this step in appendix E.

## 3 Experiments

We empirically evaluate GEOMOL on the task of low-energy conformer ensemble generation for small and drug-like molecules. We largely follow the evaluation protocols of recent methods [Simm and Hernandez-Lobato, 2020, Xu et al., 2021], but also introduce new useful metrics.

**Datasets & splits.** We use two popular datasets: GEOM-QM9 [Ramakrishnan et al., 2014] and GEOM-DRUGS [Axelrod and Gomez-Bombarelli, 2020a]. Statistics and other details are in fig. 10 and in Mansimov et al. [2019]. Datasets are preprocessed as described in appendix G. We split them randomly based on molecules into train/validation/test (80%/10%/10%). At the end, for each dataset, we sample 1000 random test molecules as the final test set. Thus, the splits contain 106586/13323/1000 and 243473/30433/1000 molecules for GEOM-QM9 and GEOM-DRUGS, resp.

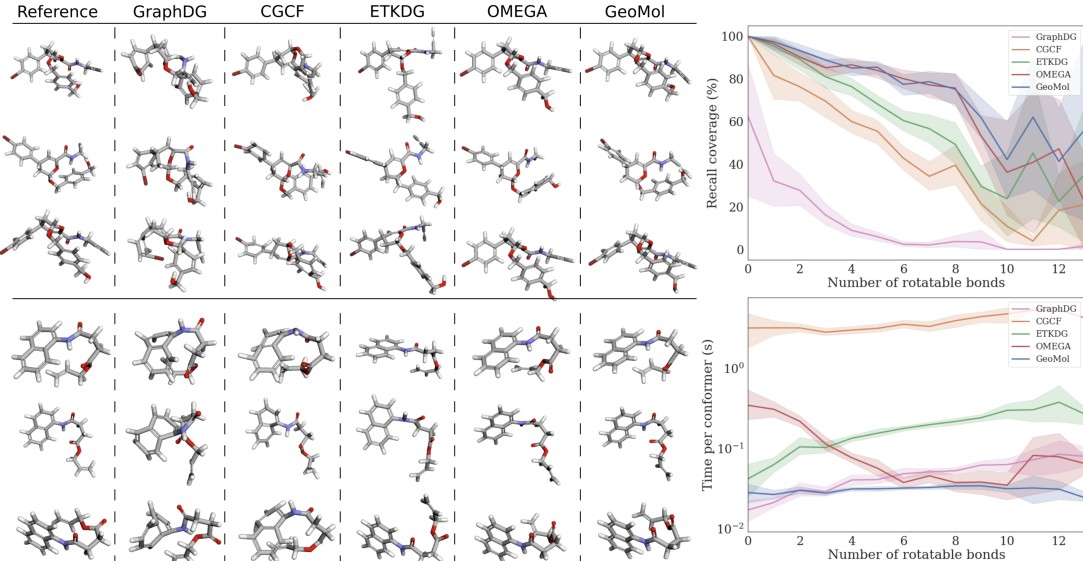

Figure 7: Left: Examples of generated structures. For every model, we show the best generated conformer, i.e., with the smallest RMSD to the shown ground truth. More examples are in appendix N. Right/top: Number of rotatable bonds per DRUGS test molecule versus COV Recall (95% confidence intervals). Right/bottom: conformer generation times for each model.

**Baselines.** We compare to established or recent baselines (discussed in section 1). **ETKDG/RDKit** [Riniker and Landrum, 2015] is likely the most popular open-source software, a stochastic DG-based method developed in the RDKit package. **OMEGA** [Hawkins et al., 2010, Hawkins and Nicholls, 2012, Friedrich et al., 2017], a rule-based method, is one of the most established commercial software, with more than a decade of continuous development. OMEGA and ETKDG are some of the fastest and best scaling existing approaches. Finally, we compare with the recent ML models of highest reported quality: **GraphDG** [Simm and Hernandez-Lobato, 2020] and **CGCF** [Xu et al., 2021].

**Evaluation metrics.** We follow prior work [Simm and Hernandez-Lobato, 2020, Xu et al., 2021] and use root-mean-square deviation of atomic positions (RMSD) to compare any two conformers. This is defined as the normalized Frobenius norm of the two corresponding matrices of 3D coordinates after being SE(3)-aligned a priori (using the Kabsch alignment algorithm [Kabsch, 1976]). Next, we introduce four types of metrics to compare two conformer ensembles, generated by a method, $\{\mathcal{C}_k\}_{k\in[1..K]}$, and ground truth, $\{\mathcal{C}_l^*\}_{l\in[1..L]}$. These metrics follow the established classification metrics of Precision and Recall and are defined for a given threshold $\delta > 0$ as:

$$\text{COV - R (Recall)} \stackrel{\text{def}}{=} \frac{1}{L}\left|\{l \in [1..L] : \exists k \in [1..K], RMSD(\mathcal{C}_k, \mathcal{C}_l^*) < \delta\}\right|$$

$$\text{AMR - R (Recall)} \stackrel{\text{def}}{=} \frac{1}{L}\sum_{l\in[1..L]} \min_{k\in[1..K]} RMSD(\mathcal{C}_k, \mathcal{C}_l^*)$$

(5)

where AMR is "Average Minimum RMSD", COV is "Coverage", and *COV - P (Precision)* and *AMR - P (Precision)* are defined as in eq. (5), but with the generated and ground truth conformer sets swapped. The recall metrics measure how many of the ground truth conformers are correctly predicted, while the precision metrics indicate how many generated structures are of high quality. Specifically, in terms of recall, COV measures the percentage of correct generated conformers from the ground truth set (where a correct conformer is defined as one within an RMSD threshold of the true conformer), while AMR measures the average RMSD of each generated conformer with its closest groun truth match. Depending on the application, either of the metrics might be of greater interest. We follow Xu et al. [2021] and set $\delta = 0.5$Å for GEOM-QM9 and $\delta = 1.25$Å for GEOM-DRUGS.

**Training and test details.** For each input molecule having $K$ ground truth conformers, we generate exactly $2K$ conformers using any of the considered methods. For GEOMOL, this is done by sampling

different random noise vectors that are appended to node and edge features before the MPNN (eq. (1)). At train time, our model uses a standard deviation (std) $s$ (see eq. (1)) of 5 for both GEOM-QM9 and GEOM-DRUGS. At test time, GEOMOL can use the same or different $s$ values, depending on the downstream application, i.e., higher $s$ results in more diverse conformers, while lower $s$ gives more quality (better precision). For OMEGA, it is not possible to specify a desired number of conformers. So, we tune the RMSD threshold (which decides how many conformers to keep) such that the total generated conformers by OMEGA are approximately $2K$. For GEOM-QM9, this corresponds to no RMSD cutoff (i.e., OMEGA generates all possible conformers), and for GEOM-DRUGS, this corresponds to a cutoff of 0.7Å (meaning no two generated conformers will have a distance smaller than this cutoff). We discuss hyper-parameters and additional training details in appendix H.

**Results & discussion.** Results are shown in table 1 and table 2, and confidence intervals are in appendix I. As noted above, GEOMOL can be run with different noise std at test time, depending on which metric the user is interested in. Even though OMEGA is an established commercial software with more than a decade of continuous development, our model remarkably frequently outperforms it. Note that OMEGA fails to generate any conformers for 7% of the QM9 test set (many of which include fused rings). Moreover, we also outperform the popular RDKit/ETKDG open-source model (except for AMR-P on QM9) and very recent ML models such as **GraphDG** and **CGCF**, sometimes by a large margin. For a qualitative insight, we show generated examples in fig. 7 and appendix N.

Additionally, we show in fig. 7 how COV Recall results are affected by the increasing number of rotatable bonds in the test molecule. As expected, having more rotatable bonds makes the problem harder, and this affects all baselines, but GEOMOL maintains a reasonable coverage even for more difficult molecules. Moreover, in appendix K and table 8 we show energy calculations of the generated conformers to support their plausibility. Additionally, results with energy-based relaxations are given in appendix J.

**Running time.** Fig. 7 shows conformer generation test running times. Our model is the fastest method from the considered baselines, being much faster than **CGCF** or **ETKDG/RDKit**. Moreover, GEOMOL scales favorably for molecules with increasing number of rotatable bonds.

# 4   Conclusion

We proposed GEOMOL, an end-to-end generative approach for molecular 3D conformation ensembles that explicitly models various molecular geometric aspects such as torsion angles or chirality. We expect that such differentiable structure generators will significantly impact small molecule conformer generation along with many related applications (e.g., protein-ligand binding), thus speeding up areas such as drug discovery. GEOMOL's full source code will be made publicly available.

**Limitations & future work.** A few current limitations are highlighted and left for future extensions (see also discussion in appendix O). First, our model does not currently support disconnected molecular graphs, e.g., ionic salts, but it can be applied to each connected component, followed by a 3D alignment. Next, our approach would benefit from explicit modeling of long distance interactions, especially for macrocycles or large molecules. This remains to be addressed in an efficient manner. Third, explicitly using ground truth energy values could further improve GEOMOL. Last, we look forward to fine-tune GEOMOL on applications such as generating molecular docking poses or descriptors for 4D QSAR.

## Acknowledgments and Disclosure of Funding

OEG thanks Bracha Laufer, Tian Xie, Xiang Fu, Peter Mikhael, and the rest of the RB and TJ group members for their helpful comments and suggestions. LP thanks Camille Bilodeau and the rest of the WHG, KFJ, and CWC research groups for their useful discussions. We also thank Pat Walters, Simon Axelrod, and Rafael Gomez-Bombarelli for their insightful feedback as well as Minkai Xu and Shitong Luo for helping run the CGCF model. Both OEG and LP are funded by the Machine Learning for Pharmaceutical Discovery and Synthesis consortium.

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
