# A  Proof of proposition 1

*Proof.* Let's assume we apply a random CCW torsion rotation of angle $\gamma \in (0, 2\pi)$ around bond XY to all bonds $XT_i$. We will prove that the resulting $\alpha$ will be shifted exactly by $\gamma$. Let us denote the current dihedrals after the $\gamma$ rotation by $\Delta_{ij}^{cur} = \angle^{cur}(XYT_i, XYZ_j) \bmod 2\pi$. Also, denote by $\Delta_{ij} = \angle(XYT_i, XYZ_j) \bmod 2\pi$ the dihedral angles before rotation. We have the relation $\Delta_{ij} = [\Delta_{ij}^{cur} + \gamma] \bmod 2\pi, \forall i, j$, or, in matrix form:

$$\mathbf{s}_{ij} = \mathbf{A}_\gamma \mathbf{s}_{ij}^{cur} \tag{6}$$

where $\mathbf{A}_\gamma := \begin{bmatrix} \cos(\gamma) & -\sin(\gamma) \\ \sin(\gamma) & \cos(\gamma) \end{bmatrix}$, $\mathbf{s}_{ij} := \begin{bmatrix} \cos(\Delta_{ij}) \\ \sin(\Delta_{ij}) \end{bmatrix}$ and $\mathbf{s}_{ij}^{cur} := \begin{bmatrix} \cos(\Delta_{ij}^{cur}) \\ \sin(\Delta_{ij}^{cur}) \end{bmatrix}$.

We know use $\mathbf{s} := \sum_{i,j} c_{ij} \mathbf{s}_{ij} \in \mathbb{R}^2$ and $\mathbf{s}^{cur} := \sum_{i,j} c_{ij} \mathbf{s}_{ij}^{cur} \in \mathbb{R}^2$. From the above relation, we obtain that $\frac{\mathbf{s}}{\|\mathbf{s}\|} = \mathbf{A}_\gamma \frac{\mathbf{s}^{cur}}{\|\mathbf{s}^{cur}\|}$. Writing $\frac{\mathbf{s}}{\|\mathbf{s}\|} = \begin{bmatrix} \cos(\alpha) \\ \sin(\alpha) \end{bmatrix}$ and $\frac{\mathbf{s}^{cur}}{\|\mathbf{s}^{cur}\|} = \begin{bmatrix} \cos(\alpha^{cur}) \\ \sin(\alpha^{cur}) \end{bmatrix}$ we obtain that $\alpha = \gamma + \alpha^{cur}$ which concludes our proof. $\qquad\square$

**A note on collinear 3D points.**   In the main text, we always assumed that dihedral angles $\Delta_{ij} \overset{\text{def}}{=} \angle(XYT_i, XYZ_j), \forall i \in [1..n], \forall j \in [1..m]$ exist. However, any such angle will not be defined if X and Y are collinear with either $T_i$ or $Z_j$. In such cases, we identify those neighbor nodes that are predicted to be collinear with X and Y (solely based on their respective LS), and then remove them from the computation of $\mathbf{A}^{cur}$ and $\mathbf{s}_\gamma$. In the extreme case when these collinear nodes are the only neighbors, the torsion angle is not defined by our formulation. We leave this isolated case to be solved in future extensions of our work.

# B  Initial assembly of the LSs of the endpoints of a bond XY

We detail here the formulae used in section section 2.4. Let the (predicted) LS of node X be $\mathbf{p}_X = \mathbf{0}, \mathbf{p}_Y, \mathbf{p}_{T_1}, \ldots, \mathbf{p}_{T_n} \in \mathbb{R}^3$, as in section 2.3. Similarly, let the LS of node Y be $\mathbf{q}_Y = \mathbf{0}, \mathbf{q}_X, \mathbf{q}_{Z_1}, \ldots, \mathbf{q}_{Z_m} \in \mathbb{R}^3$. By design of section 2.3, we have that $\|\mathbf{q}_X\| = \|\mathbf{p}_Y\|$. These sets of 3D points can be aligned by applying any SE(3) transformation to each of them. To achieve alignment, we first want to match the X and Y points.

For this, we first rotate the LS of X such that $\mathbf{p}_Y$ becomes $\begin{bmatrix} \|\mathbf{p}_Y\| \\ 0 \\ 0 \end{bmatrix}$ and $\mathbf{p}_X$ remains $\mathbf{0}$. Next, we rotate and translate the LS of Y such that $\mathbf{q}_Y$ becomes $\mathbf{p}_Y$ and $\mathbf{q}_X$ becomes $\mathbf{0}$. More concretely, let $\eta_X \in \mathbb{R}^3$ be any random unit-norm vector orthogonal to $\mathbf{p}_Y$. We find a rotation matrix $\mathbf{H}_{X,Y} = \begin{bmatrix} - & \mathbf{h}_1^\top & - \\ - & \mathbf{h}_2^\top & - \\ - & \mathbf{h}_3^\top & - \end{bmatrix} \in \mathbb{R}^{3\times3}$ such that $\mathbf{H}_{X,Y}\mathbf{p}_Y = \begin{bmatrix} a \\ 0 \\ 0 \end{bmatrix}$, with $a > 0$, and $\mathbf{H}_{X,Y}\eta_X = \begin{bmatrix} b \\ c \\ 0 \end{bmatrix}$, where $b, c \in \mathbb{R}$. This is solved as:

$$\mathbf{h}_1 = \frac{\mathbf{p}_Y}{\|\mathbf{p}_Y\|}; \quad \mathbf{h}_3 = \frac{\mathbf{p}_Y \times \eta_X}{\|\mathbf{p}_Y \times \eta_X\|}; \quad \mathbf{h}_2 = -(\mathbf{h}_1 \times \mathbf{h}_3)$$

We now rotate the LS of X by doing: $\mathbf{p}_{T_i}' \overset{\text{def}}{=} \mathbf{H}_{X,Y}\mathbf{p}_{T_i}, \forall i \in [1..n], \mathbf{p}_Y' \overset{\text{def}}{=} \mathbf{H}_{X,Y}\mathbf{p}_Y$. We apply a similar procedure for the LS of Y, computing $\mathbf{q}_{Z_j}' \overset{\text{def}}{=} \mathbf{H}_{Y,X}\mathbf{q}_{Z_j}, \forall j \in [1..m]$ and $\mathbf{q}_X' := \mathbf{H}_{Y,X}\mathbf{q}_X$. We now reflect the LS of Y w.r.t. the first and second coordinate by doing $\mathbf{q}_{Z_j}' \leftarrow \begin{bmatrix} -1 & 0 & 0 \\ 0 & -1 & 0 \\ 0 & 0 & 1 \end{bmatrix} \mathbf{q}_{Z_j}' + \mathbf{p}_Y'$, which centers X in the origin, i.e., $\mathbf{q}_X' = \mathbf{0}$, and aligns the two vectors of Y, i.e., $\mathbf{q}_Y' = \mathbf{p}_Y'$.

## C  Dihedral angle formula

The CCW dihedral angle between two intersecting half-planes ABC and ABD with common points A,B is computed as[6]

$$\angle(ABC, ABD) = atan2(\|\mathbf{b}_2\|\langle\mathbf{b}_1, \mathbf{b}_2 \times \mathbf{b}_3\rangle, \langle\mathbf{b}_1 \times \mathbf{b}_2, \mathbf{b}_2 \times \mathbf{b}_3\rangle) \tag{7}$$

where $\mathbf{b}_1 := \mathbf{s}_A - \mathbf{s}_C, \mathbf{b}_2 := \mathbf{s}_B - \mathbf{s}_A, \mathbf{b}_3 := \mathbf{s}_D - \mathbf{s}_B$.

## D  Details of the loss function

Assume, for a molecular graph $G = (V, E)$, that one predicted/generated conformer using one model is $\mathcal{C}$, while the corresponding ground truth conformer is $\mathcal{C}^*$. We use notations in section 2. $E$ is the set of edges in graph $G$. For every chain of 3 edges $(u, v), (v, w), (w, y) \in E$, we define $\angle(uvwy) \stackrel{\text{def}}{=} \angle(uvw, vwy)$.

Similar to AlphaFold [Senior et al., 2020], we fit distances using normal distributions and angles using von Mises distributions. The resulting negative log-likelihood loss averages over same types of terms, being formally written as:

$$
\begin{aligned}
\mathcal{L}(\mathcal{C}, \mathcal{C}^*) \stackrel{\text{def}}{=} & \xi_1 \cdot \frac{1}{\#\{(u, v) \in E\}} \sum_{\{(u,v)\in E\}} (d(u, v) - d^*(u, v))^2 \\
&+ \xi_2 \cdot \frac{1}{\#\{u, v : \text{2-hops away}\}} \sum_{\{u,v:\text{2-hops away}\}} (d(u, v) - d^*(u, v))^2 \\
&+ \xi_3 \cdot \frac{1}{\#\{u, v : \text{3-hops away}\}} \sum_{\{u,v:\text{3-hops away}\}} (d(u, v) - d^*(u, v))^2 \\
&- \xi_4 \cdot \frac{1}{\#(u, v) \in E, (v, w) \in E} \sum_{(u,v)\in E,(v,w)\in E} \cos(\angle uvw - \angle^* uvw) \\
&- \xi_5 \cdot \frac{1}{\#(u, v), (v, w), (w, y) \in E} \sum_{(u,v),(v,w),(w,y)\in E} \cos(\angle(uvwy) - \angle^*(uvwy))
\end{aligned}
\tag{8}
$$

where $\xi_1, \xi_2, \xi_3, \xi_4, \xi_5 > 0$ are hyperparameters tuned on the validation set, with $\xi_1, \xi_2, \xi_3$ representing inverse standard deviations of the respective normal distributions, while $\xi_4, \xi_5$ being the measures of concentration of the corresponding von Mises distributions.

**Dealing with node symmetries. A matching loss function.**  So far, we haven't tackled the following difficulty: symmetric graph nodes that are less than 3 hops away are indistinguishable by MPNNs in general, and by our current model in particular, even if we append initial random noise feature vectors. However, these nodes have distinct distances and angles to other nodes in 3D space. Examples are hydrogen groups as in fig. 8. The difficulty in our case comes from the fact that the MPNN has lost the true matching between the graph nodes and the respective 3D points of the ground truth conformer. Our model is able to differentiate symmetric nodes because of the initial random noise vectors appended to each atom and feature vector, but a consistent matching with the ground truth conformer nodes is lost, and we seek to recover it.

We here address this problem only for symmetric terminal nodes attached to the same common neighbor. We leave

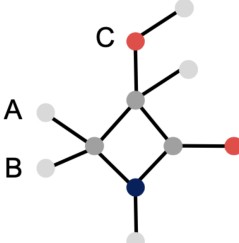

Figure 8: Nodes A and B are symmetric in this example graph, but not when placed in the 3D space, as node A will be spatially closer to node C than node B. Such cases require a special treatment.

---

[6]See https://en.wikipedia.org/wiki/Dihedral_angle.

further extensions to more general symmetries for future work. Concretely, let $X_1, \ldots, X_T$ be identical atoms of degree 1, each being connected to the same common node $Y$ in the molecular graph. Let $\mathbf{c}_1, \ldots, \mathbf{c}_T \in \mathbb{R}^3$ be their predicted 3D coordinates in one generated conformer $\mathcal{C} : V \to \mathbb{R}^3$. Let also $\mathbf{c}_1^*, \ldots, \mathbf{c}_T^* \in \mathbb{R}^3$ be their corresponding 3D coordinates of some ground truth conformer $\mathcal{C}^* : V \to \mathbb{R}^3$. These coordinates are then used for the computation of different loss terms in eq. (8).

We propose a new loss function based on eq. (8) that replaces it:

$$\mathcal{L}^{perm}(\mathcal{C}, \mathcal{C}^*) \stackrel{\text{def}}{=} \min_{\pi \in S_T} \mathcal{L}(\mathcal{C}, \mathcal{C}_\pi^*) \tag{9}$$

where $\mathcal{C}_\pi^* : V \to \mathbb{R}^3$ is defined as $\mathcal{C}_\pi^*(v) = \mathcal{C}^*(v), \forall v \in V \setminus \{X_1, \ldots, X_T\}$ and $\mathcal{C}_\pi^*(X_i) = \mathcal{C}^*(X_{\pi(i)}), \forall i \in [1..T]$. In a nutshell, this new loss function searches all possible permutations of the symmetric nodes $X_1, \ldots, X_T$ in order to find the "best matching" (based on the loss value) with the corresponding points in the ground truth conformer $\mathcal{C}^*$. The procedure above is applied for all groups of symmetric graph nodes that are of degree 1 and attached to a common node in the molecular graph. The loss in eq. (9) does not significantly affect the speed of our method (as we show in the experimental section) due to the fact that most groups of symmetric atoms contain at most 3 nodes. However, it positively affects the quality of the resulting conformers as exemplified in fig. 6.

**OT Loss.** During training, the matrix $\mathbf{T}$ in section 2.5 is computed for the forward pass using the Earth Mover Distance (EMD) function from the POT library [Flamary and Courty, 2017] and kept fixed during backpropagation. The EMD computation cannot be parallelized in mini-batches in the current version of the library, but everything else is batch-parallelizable in our model (e.g., computation of $\mathcal{L}(\mathcal{C}, \mathcal{C}^*)$).

# E   Details of the full conformer assembly procedure at test time

The training stage happens without assembling the full conformer. However, we desire to generate the full 3D conformer for test and inference time. This can be done by repeatedly applying the assembly operation described in section 2.4. We assemble the 3D coordinates of the atoms one by one, following a fixed arbitrary graph traversal, say Breadth First Search (BFS). Our procedure described here is deterministic, and any graph traversal will lead to the same final conformer (up to an SE(3) transformation).

The key step is assembling two sets of 3D points: the set containing the LS of node X (and possibly other atom coordinates added in previous steps), denoted as $S_X := \{\mathbf{p}_1, \ldots, \mathbf{p}_n\} \subset \mathbb{R}^3$, and the set containing the LS of node Y, denoted as $S_Y := \{\mathbf{q}_1, \ldots, \mathbf{q}_m\} \subset \mathbb{R}^3$. Assume that X and Y are connected by a bond/edge. Also assume that X is connected to nodes $T_i$, while $Y$ is connected to nodes $Z_j$ as denoted in fig. 5. We make the clarification that $\mathbf{p}_{T_i} \in S_X, \mathbf{p}_Y \in S_X, \mathbf{p}_X \in S_X$ and $\mathbf{q}_{Z_j} \in S_Y, \mathbf{q}_Y \in S_Y, \mathbf{q}_X \in S_Y$ and we expect that $\|\mathbf{p}_X - \mathbf{p}_Y\| = \|\mathbf{q}_X - \mathbf{q}_Y\|$ by our design of this model.

To assemble the sets $S_X$ and $S_Y$, we follow the steps described in section 2.4 and appendix B, but using only points in the LS of $X$ from the set $S_X$ together with only points of the LS of $Y$ from $S_Y$ to compute the torsion angle $\gamma$ of the bond $XY$. That is, the torsion angle is computed only based on the graph neighbors of nodes X and Y. However, all points in $S_X$ and $S_Y$ are used for all the updates of the 3D coordinates. In the end, we obtain assembled set of points that satisfy

$$\mathbf{q}_Y = \mathbf{p}_Y = \begin{bmatrix} d_{GNN}(\mathbf{h}_X, \mathbf{h}_Y) \\ 0 \\ 0 \end{bmatrix}, \mathbf{q}_X = \mathbf{p}_X = \mathbf{0}.$$ Finally, we just merge the two sets $S_X$ and $S_Y$

into their union.

**Dealing with graph cycles.** The assembly operation might lead to inconsistencies when a node is reached the second time after a cycle traversal. We adopt a deterministic approach to tackle this issue. We first fix one BFS traversal of the graph and assemble the LS in the order given by this traversal. If the current node is not part of a cycle, then we can perform the assembling as described before. However, when we first encounter a node that is part of a cycle of nodes $X_1, X_2, \ldots, X_n$, we will jointly compute all the 3D coordinates of this cycle and attach the entire ring structure to the current partial conformer. Concretely, for every $i \in \{1, \ldots, n\}$, we assemble the LS of nodes

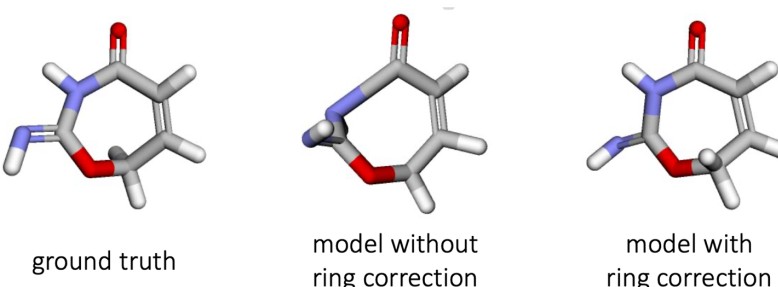

ground truth      model without ring correction      model with ring correction

Figure 9: Effect of two assembly techniques. Left: ground truth structure. Middle: our assembly procedure is applied on a random spanning tree of the molecular graph. Right: our described ring correction/smoothing algorithm, leading to a visibly improved structure.

$i, i+1, \ldots n, 1, 2, \ldots i-2$ resulting in the 3D points $\mathbf{x}_{i-1}, \mathbf{x}_i \mathbf{x}_{i+1}, \ldots, \mathbf{x}_n, \mathbf{x}_1, \mathbf{x}_2, \ldots, \mathbf{x}_{i-2}, \mathbf{x}'_{i-1}$, where node $i-1$ has two 3D points $\mathbf{x}_{i-1}$ and $\mathbf{x}'_{i-1}$ computed based on the LS of nodes $i$ and $i-2$ respectively. Next, we average the two 3D points of node $i-1$ into a single vector denoted as $\mathbf{x}_{i-1} \in \mathbb{R}^3$. We denote the resulting list of 3D points as $S_i = \{\mathbf{x}_{i-1}, \mathbf{x}_i \mathbf{x}_{i+1}, \ldots, \mathbf{x}_n, \mathbf{x}_1, \mathbf{x}_2, \ldots, \mathbf{x}_{i-2}\}$. So, we have obtained $n$ lists of 3D points, $S_i, \forall i \in [1..n]$, that each represents the 3D conformer of the same cycle when the assembling is done using all nodes except one, in turn. We align all $S_i$ together using Kabsch algorithm [Kabsch, 1976][7], followed by averaging all vectors of the same cycle node, for each node in the cycle, thus obtaining a final smoothed set of 3D coordinates for this respective cycle. We call this procedure "ring correction" or "ring smoothing". We additionally note that, when aligning two sets $S_i$ and $S_j$, we also have to align the non-cycle points previously assembled. Finally, after all 3D vectors of a cycle are obtained, we continue to assemble in the order given by the BFS traversal. We show an example of the effect of this procedure in fig. 9.

# F   Datasets statistics

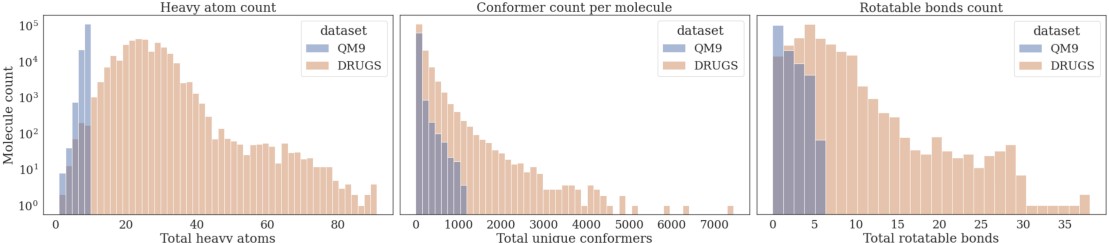

Figure 10: Datasets statistics.

We here show several dataset statistics in fig. 10, but point the interested reader to other resources describing these datasets, e.g., Mansimov et al. [2019], Axelrod and Gomez-Bombarelli [2020a], Ramakrishnan et al. [2014].

**Datasets generation details.** The GEOM dataset is generated with semi-empirical tight-binding DFT (GFN2-xTB) with the CREST software Grimme [2019]. However, such semi-empirical calculations can be relatively slow in practice–espeically for larger molecules–which is one of the main motivations of ML models for conformer ensemble prediction.

---

[7]https://en.wikipedia.org/wiki/Kabsch_algorithm

# G    Data preprocessing

We featurize each .pickle file in the datasets, which correspond to a single molecule and its respective conformers, to PyTorch Geometric data objects. First, we ensure the provided SMILES strings can be processed by RDKit, and we discard the few molecules that fail to meet this criteria. We additionally discard molecules with disconnected fragments (i.e., contains a "." in the SMILES) and without any dihedral angles (i.e., fails to match a "$[*] \sim [*] \sim [*] \sim [*]$" SMARTS pattern). Finally, some conformers in the original dataset may have reacted during the original data generation process. We filter these conformers by inferring the SMILES from the 3D structure, canonicalizing the SMILES with RDKit, and checking this SMILES with the SMILES reported by the dataset. Any conformers who fail this check are discarded as reacted conformers. We detail the atom and bond features used for the final GEOM-DRUGS model in Table 3. Note that the GEOM-QM9 model uses the same features but restricts the possible atom identities to H, C, N, O, and F.

Table 3: Atom and bond features

| Atom features | | | |
|---|---|---|---|
| Indices | Description | Options | Type |
| 0-34 | atom identity | H, Li, B, C, N, O, F, Na, Mg, Al, Si, P, S, Cl, K, Ca, V, Cr, Mn, Cu, Zn, Ga, Ge, As, Se, Br, Ag, In, Sb, I, Gd, Pt, Au, Hg, Bi | one-hot |
| 35 | atomic number | $\mathbb{Z}_{>0}$ | value |
| 36 | aromaticity | true, false | one-hot |
| 37-44 | degree | 0, 1, 2, 3, 4, 5, 6, other | one-hot |
| 45-50 | hybridization | $sp$, $sp^2$, $sp^3$, $sp^3d$, $sp^3d^2$, other | one-hot |
| 51-58 | implicit valence | 0, 1, 2, 3, 4, 5, 6, other | one-hot |
| 59-62 | formal charge | -1, 0, 1, other | one-hot |
| 63-69 | presence of atom in ring of size x | 3, 4, 5, 6, 7, 8, other | k-hot |
| 70-74 | number of rings atom is in | 0, 1, 2, 3, other | one-hot |

| Bond features | | | |
|---|---|---|---|
| Indices | Description | Options | Type |
| 0-3 | bond type | single, double, triple, aromatic | one-hot |

Before computing performance statistics (coverage and AMR), we attempt to clean the ground truth data. During the original CREST simulations to generate the GEOM dataset, some of the conformers may have undergone reactions (e.g., dissociation), which leads to a different molecular graph than desired. We remove such conformer from the ground truth set before computing statistics. For GEOMOL, we must define an input SMILES to generate structures, and we must accurately define all existing stereocenters. In some molecules with defined E/Z stereochemistry, the conformers from the true dataset flip between E and Z so often that we remove the defied stereochemistry from the input SMILES to GEOMOL. There are also cases when stereochemsitry exists in a SMILES even when it is not explicitly defined (e.g., with fused rings, where the alternate stereochemistry would lead to a nonsensical structure, so it does not need to be defined). Defining these stereocenters is crucial for the performance of our model, although we realize this is a drawback for some practical use cases. Future studies should automate the labeling of such stereocenters.

# H    Additional training details and hyperparameters

We run each model for a fixed 250 epochs with early stopping and keep the model with the best validation performance. This strategy determines the final hyperparameters chosen for the model, shown in table 4. We additionally tune the standard deviation $s$ on a sample of the validation set to maximize recall coverage or precision coverage (depending on the user's interest), and we show the corresponding test scores in our tables. To adjust the learning rate during training, we use a plateau scheduler which automatically decays the learning rate when improvement has stalled, using

Table 4: Hyerparameter choices

| Hyperparameter | Final choice (QM9/DRUGS) |
|---|---|
| True conformers sampled | 10/20 |
| Model conformers generated | 10/20 |
| Model dimension | 25 |
| Random vector dimension | 10 |
| Random vector standard deviation $s$ | 5 |
| MPNN 1 depth | 3 |
| MPNN 1 number of layers | 2 |
| MPNN 2 depth | 3 |
| MPNN 2 number of layers | 2 |
| LS self-attention encoder heads | 2 |
| LS coordinate MLP layers | 2 |
| LS distance MLP layers | 1 |
| $h_{mol}$ MLP layers | 1 |
| $\alpha$ MLP layers | 2 |
| $c_{ij}$ MLP layers | 1 |
| Batch size | 16 |
| Learning rate | 1e-3 |

Table 5: Confidence intervals for GEOMOL with the default noise $s = 5$.

| Models | COV - R (%) ↑ | | AMR - R (Å) ↓ | | COV - P (%) ↑ | | AMR - P (Å) ↓ | |
|---|---|---|---|---|---|---|---|---|
| | Mean | Median | Mean | Median | Mean | Median | Mean | Median |
| GEOMOL DRUGS | 82.43 ± 0.26 | 95.10 ± 0.25 | 0.8626 ± 0.0020 | 0.8374 ± 0.0037 | 78.52 ± 0.02 | 94.40 ± 0.20 | 0.9336 ± 0.0015 | 0.8567 ± 0.0030 |
| GEOMOL QM9 | 91.52 ± 0.36 | 100.00 ± 0.00 | 0.2254 ± 0.0012 | 0.1938 ± 0.0024 | 86.71 ± 0.29 | 100.00 ± 0.00 | 0.2702 ± 0.0007 | 0.2413 ± 0.0030 |

a decay factor of 0.7 and a patience of 5 epochs. All models use the same seed to initialize network weights, ensuring consistency between runs. During training time, we randomly sample a fixed number of ground truth conformers from each molecule. This fixed number is a hyperparameter in our optimization, which we set to 10 for GEOM-QM9 and 20 for GEOM-DRUGS. This value is not relevant during test time, as the model simply generates the number of conformers requested by the user.

## I  Confidence intervals

We report in table 5 confidence intervals for GEOMOL on both datasets during test time. The same trained model is run for 3 times for a given test molecule with K ground truth conformers, where one run means generating $2K$ conformers.

## J  Results with force field optimization

We also show results in table 6 and table 7 for all models after additional fine-tuning with FF energy minimization. We used MMFF94s Merck Molecular Force Field [Halgren, 1996, 1999]). Note that the best GEOMOL models here use $s = 1$ during training rather than $s = 5$.

## K  Energy calculations

To gauge the plausibility of generated conformers, we compute the energies as defined by the MMFF force field within RDKit for conformers generated with ML-based methods before force field fine tuning. We note that an improved demonstration would calculate energies at the GFN2-xTB semiempirical level of theory, since the training data were generated with this method, but we decided to use a force field instead, which should retain the qualitative trends. Table 8 illustrates these results,

Table 6: Results on the **GEOM-DRUGS** dataset with FF fine-tuning.

| Models | COV - R (%) ↑ | | AMR - R (Å) ↓ | | COV - P (%) ↑ | | AMR - P (Å) ↓ | |
|---|---|---|---|---|---|---|---|---|
| | Mean | Median | Mean | Median | Mean | Median | Mean | Median |
| GraphDG *(ML)* | 85.33 | 100.00 | 0.859 | 0.831 | 62.65 | 69.45 | 1.162 | 1.121 |
| CGCF *(ML)* | **91.25** | **100.00** | 0.723 | 0.702 | 63.86 | 70.43 | 1.096 | 1.055 |
| RDKit/ETKDG | 77.22 | 87.62 | 0.886 | 0.837 | 75.98 | 93.86 | 0.911 | 0.803 |
| OMEGA *(C)* | 80.82 | 94.74 | 0.840 | 0.755 | 81.79 | **100.00** | 0.804 | 0.684 |
| GEOMOL ($s = 0.25$) | 80.73 | 92.01 | 0.863 | 0.809 | **86.12** | **100.00** | **0.751** | **0.616** |
| GEOMOL ($s = 3.00$) | **91.34** | **100.00** | **0.683** | **0.663** | 79.64 | 92.46 | 0.841 | 0.756 |

Table 7: Results on the **GEOM-QM9** dataset with FF fine-tuning.

| Models | COV - R (%) ↑ | | AMR - R (Å) ↓ | | COV - P (%) ↑ | | AMR - P (Å) ↓ | |
|---|---|---|---|---|---|---|---|---|
| | Mean | Median | Mean | Median | Mean | Median | Mean | Median |
| GraphDG *(ML)* | 88.70 | 100.00 | 0.210 | 0.165 | 90.14 | **100.00** | 0.185 | 0.129 |
| CGCF *(ML)* | 75.45 | 100.00 | 0.313 | 0.246 | 50.29 | 50.00 | 0.518 | 0.520 |
| RDKit/ETKDG | 83.48 | 100.00 | 0.219 | 0.172 | 89.78 | **100.00** | **0.160** | **0.116** |
| OMEGA *(C)* | 85.73 | 100.00 | **0.177** | **0.126** | 83.08 | **100.00** | 0.224 | 0.186 |
| GEOMOL ($s = 5.00$) | **89.37** | 100.00 | 0.201 | 0.157 | **91.92** | **100.00** | 0.173 | 0.124 |

which support the results from table 1 and table 2. The energy values from GeoMol are the lowest among the ML methods, indicating greater stability of generated conformers, especially for the druglike molecules.

## L  Compute details

We train GEOMOL using a single Nvidia Volta V100 GPU and eight Intel Xeon Gold 6248 CPUs. The exact number of hours depends on the cluster load and on early stopping, but training GEOMOL for the full 250 epochs takes approximately 1.8 days for the QM9 dataset and 6.1 days for the DRUGS dataset.

## M  Additional coverage results

We additionally show how the Recall Coverage varies with the threshold $\delta$ in fig. 11. Larger thresholds allow small errors, but penalize large errors, while smaller thresholds penalize both small and large errors.

## N  Additional examples

We show additional examples of generated conformers from the GEOM-QM9 test set in fig. 12 and from the GEOM-DRUGS test set in fig. 13.

Table 8: MMFF energy calculations for ML conformer generators

| Model | Mean/Median (kcal/mol) | |
|---|---|---|
| | QM9 | DRUGS |
| GeoMol | 17.45/17.56 | 83.72/77.52 |
| GraphDG | 26.32/24.44 | 64350.58/61732.49 |
| CGCF | 86.71/87.02 | 1380.82/556.78 |

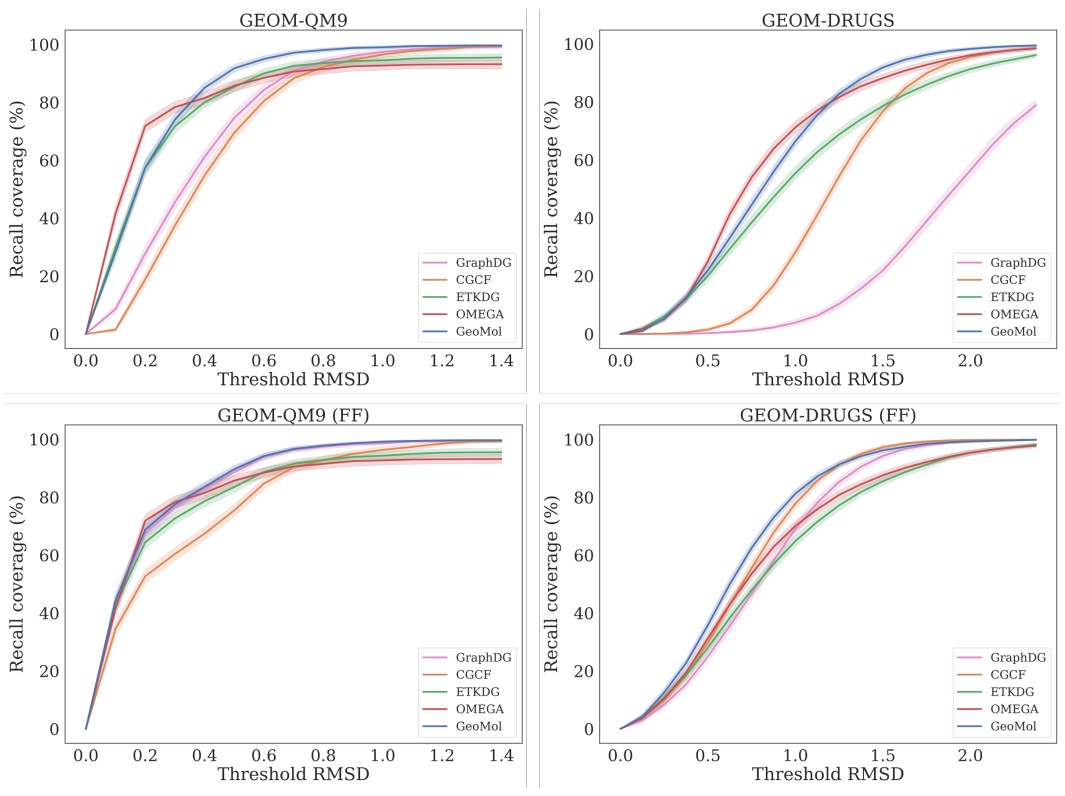

Figure 11: Recall coverage plotted against varying RMSD thresholds.

# O    Additional discussion

## O.1    Purpose of conformer generation

The purpose of conformer generation in the context of this work warrants some discussion. As stated in the main text, this purpose is heavily application-dependent. The current method is very helpful for computing the properties of drug-like molecules at low temperature, where it is necessary to generate a set of conformers that represents those with high Boltzmann populations. It can also be used for small molecules, but if the molecules are very small, it may be possible to identify all the important conformers by exhaustive search. At very high temperatures, where a large number of conformers are energetically accessible, it may be more convenient to represent the potential energy surface as a sum of hindered rotor potentials. Yet another common application for 3D conformer generation is the discovery of docked poses and 4D QSAR, where speed of the conformer generator becomes a crucial factor for virtually screening millions of structures. Crystal structure prediction (CSP) similarly relies on screening stable conformations, with recent models placing a greater emphasis on quality over quantity. These last two applications require modelling conformational flexibility rather than finding wells in the potential energy surface, as docked and crystallized conformations can fall outside these wells, carrying additional strain. The methods we present in this work can address any of these latter applications as well. We emphasize that the datasets on which GEOMOL (or any ML model) is trained guide the downstream usage, but cleverly finetuning GEOMOL on various downstream applications is an exciting future research direction.

## O.2    Comment on evaluation metrics

Additionally, we note that while the evaluation metrics presented in this work are a sensible way to benchmark ML conformer generators against each other, they do not necessarily present a realistic evaluation of these algorithms in a production setting. This is why evaluating against the OMEGA software, which does not allow the user to request a specific number of conformers, is difficult.

| Reference | GraphDG | CGCF | ETKDG | OMEGA | GeoMol |
|-----------|---------|------|-------|-------|--------|

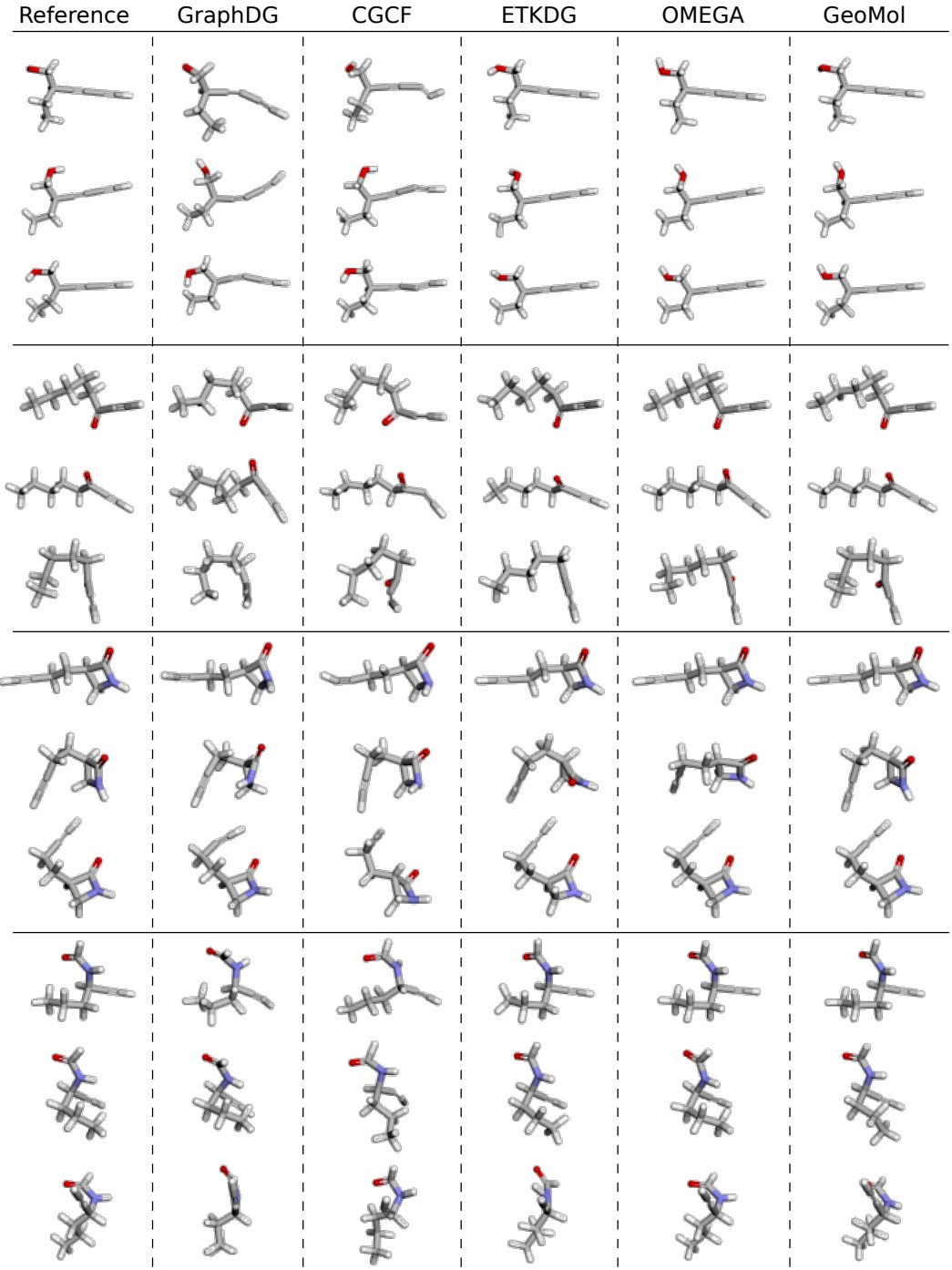

Figure 12: Additional examples of generated conformers for the GEOM-QM9 dataset.

| Reference | GraphDG | CGCF | ETKDG | OMEGA | GeoMol |
|-----------|---------|------|-------|-------|--------|

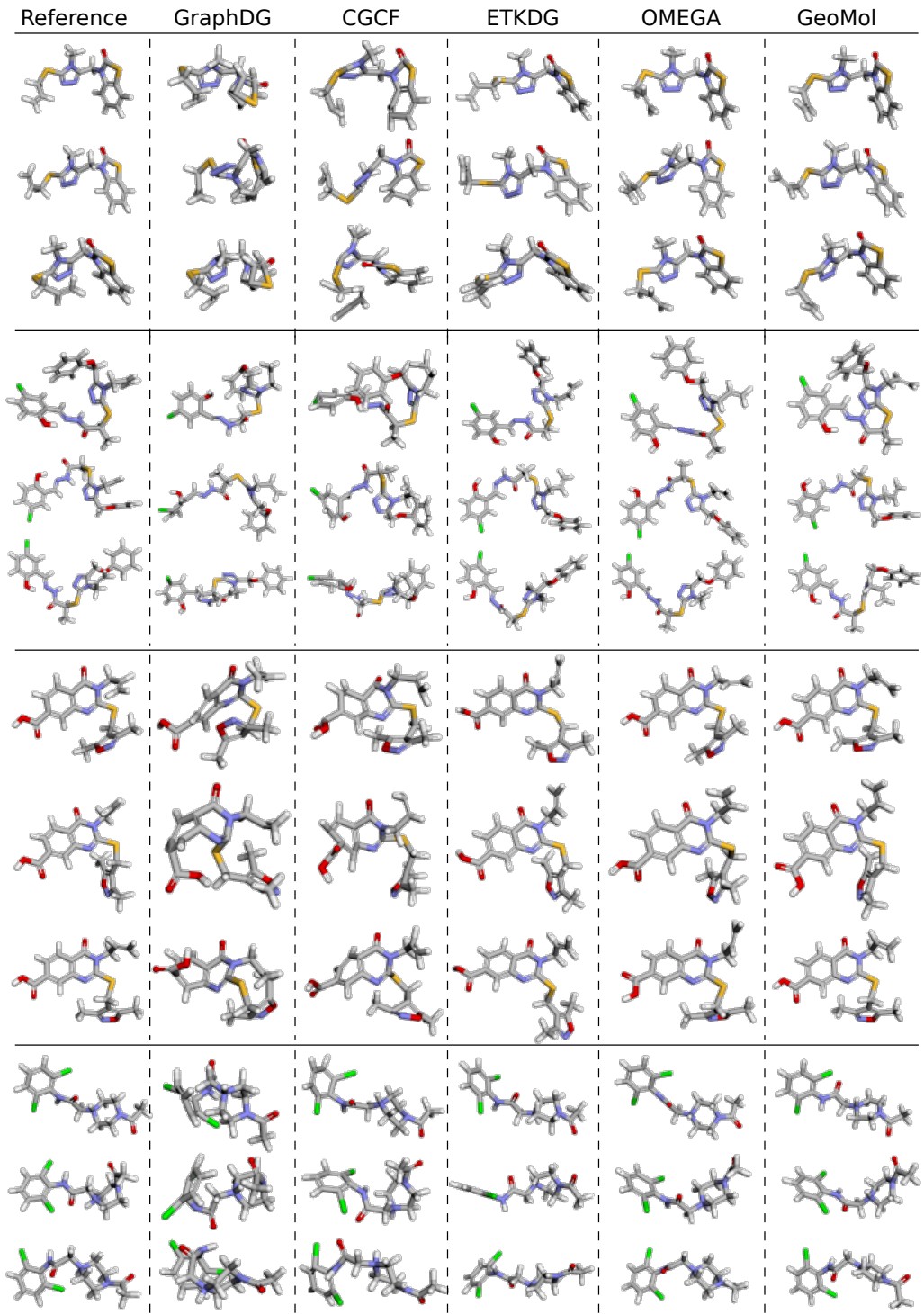

Figure 13: Additional examples of generated conformers for the GEOM-DRUGS dataset.

Further, for many applications, the generated conformers should be pruned and ranked (e.g., ranking conformers by energy and stable packing arrangements for CSP), and different conformer generation methods have varying methods to perform these post-processing steps. Because the scope of the presented work aims to distinguish our model against the many emerging ML-based conformer generators, we leave the exploration of such post-processing algorithms and strategies for future work. Furthermore, both the ETKDG and OMEGA algorithms are optimized to reproduce Protein Data Bank/Cambridge Structural Database conformations rather than replicate xTB low-energy conformations. In this sense, the evaluation presented here should not be viewed as a comprehensive argument for or against any of these methods.

## O.3 QM9 results

Focusing on the QM9 results, we see both the power and drawbacks of the popular conformer generators such as OMEGA and ETKDG. Both methods can successfully generate diverse sets of conformers for the small molecules in the QM9 dataset, as shown by the high recall coverage values. However, both methods run into similar issues. That is, they both fail to generate the requested number of conformers for a few structures in the QM9 dataset, most of which have fused rings and multiple chiral centers specified. Since OMEGA, which constructs initial 3D structures from its database of fragments, does not have many of these fragments readily available, both OMEGA and ETKDG likely run into the same issue: using DG-based methods to generate initial 3D conformers for molecules with multiple chiral centers is difficult because of the implicit chiral constraints that must be satisfied. Take for example the following SMILES, which represents a fairly simple fused ring structure: 'C#C[C@@]1(C)[C@H]2C[C@]1(C)C2'. ETKDG fails to embed a single conformer for this structure with repeated attempts. Removing the chiral centers specifications (i.e., inputting 'C#CC1(C)C2CC1(C)C2' as the SMILES) results in a successful structure generation, but many of the outputs have the incorrect chirality. While these can sometimes be corrected with energy-based minimization, the key issue of determining chiral centers exactly with DG-based methods remains. GEOMOL specifically tackles this problem such that all specified chiral centers are solved exactly (see section 2.3).

## O.4 DRUGS results

The results for the DRUGS half of GEOM dataset suggest additional issues with traditional conformer generation methods. With these larger, drug-like molecules with many rotatable bonds, the search space of conformers grows exponentially. Here, it is instructive to separate the performance of ETKDG from OMEGA. ETKDG relies on randomly sampling the space of possible interatomic distances, which leads to poor coverage for conformers where this space is large. If the goal is to improve coverage, ETKDG could benefit from improved sampling strategies. OMEGA systematically varies all torsion angles in the structure to generate ensembles, but since this process can become prohibitively expensive, we must include an RMSD cutoff for many of these input molecules so as to return results in a reasonable period of time. Furthermore, not only does OMEGA uses a modified MMFF94 FF to generate its fragment library, it also uses a FF to score each potential conformer; its identified low-energy structures will be different than the xTB geometries we use as ground truths. This is a limitation with the OMEGA algorithm as currently constructed, and it could benefit from higher quality fragment structures and improved methods to score output conformers. This is not to understate the power of OMEGA as a conformer generation tool; it significantly outperforms the other baselines in our study and should be viewed as state-of-the-art in reference.

## O.5 Runtime comparison

While the baselines ML methods have not benchmarked the speed of their algorithms, we believe this is a crucial evaluation of any conformer generator, especially if the intent is to use such a model for pose prediction and high throughput virtual screening. For each baseline, we request a fixed number of conformers and record the time necessary to output a single conformer, parallelizing each model over 40 CPU cores. We repeat this analysis three times for each method and plot the results in fig. 7. Importantly, we do not include the start up times for any of the methods (i.e., loading the neural network weights for the ML models). OMEGA, GraphDG, and GEOMOL show comparative performance, with ETKDG lagging slightly behind. Additionally, GEOMOL scales well for structures with an increasing number of rotatable bonds, especially compared to GraphDG and ETKDG. The

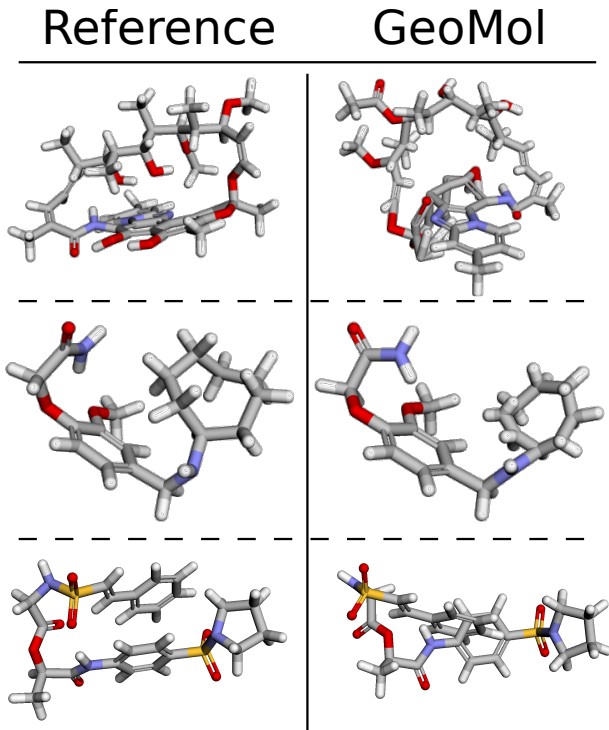

Figure 14: Common GEOMOL errors. Top: example of a macrocyle predicted poorly by GEOMOL. Middle: example of a larger ring with few training set occurrences. Bottom: example of a steric clash due to weak modeling of long-range interactions.

CGCF model is orders of magnitude slower than the other baselines, which restricts its utility for high throughput applications of conformer generation.

### O.6 Limitations

The most significant drawback of our model is its weakness in capturing long-range interactions, which can manifest itself in several ways, detailed in fig. 14. First, GEOMOL poorly models macrocycles. Because of the formulation we use to construct conformers, error accumulation sometimes leads to imperfect structures, as the ring smoothing algorithm breaks down for large cycles (top of fig. 14). We also qualitatively observe that large rings with very sparse examples in the training data can be predicted poorly by our model at test time. The middle of fig. 14 shows one such example of an eight-membered ring; here, additional training data should help, as there only exist a few eight-membered rings in the training set. Finally, the presence of steric clashes is an important issue with the current model. The bottom of fig. 14 shows one such example, being the most common error mode for GEOMOL. Since global conformer information during torsion predictions is only conveyed through the graph network representations, it is challenging to account for the position of atoms far away in graph connectivity, which can lead to overlapping atoms in the final prediction. This is especially troubling for interactions such as the pi-pi stacking depicted in the figure, which require moieties in the conformer with large 2D distances in the molecular graph to be predicted close to one another in 3D space. For this reason, extending GEOMOL with an improved method for long-range interactions is an exciting future direction.

### O.7 Conclusions

In theory, GEOMOL should replicate the quality of conformers on which it was trained, which gives the model unique power and flexibility for downstream applications. For example, one could easily construct a smaller database of high-quality conformers optimized at a higher level of theory than semi-empirical (ex. coupled-cluster) and finetune GEOMOL. These higher-quality conformers could

even include solvent effects in a range of solvents, which are often neglected in traditional conformer generation methods. Indeed, these downstream modifications could have significant impacts on a range of applications in cheminformatics and drug discovery.

Both the GraphDG and CGCF models, which represent important steps in ML-enabled conformer generation, are not quite ready to be used in a production setting without relying on a force field optimization. This is especially evident given the example structures presented in the main text and in appendix N. However, both models improve over traditional baselines at searching the space of interatomic distances and outputting diverse predictions, which is a credit to the search strategies employed by each model, especially CGCF. Further, the GraphDG model is competitive with GEOMOL for the QM9 dataset and should be viewed as a viable alternative to GEOMOL for small molecules with few heavy atoms.

As researchers continue to develop 3D ML-based conformer generators, the discussion between DG-based and non DG-based algorithms will carry on. Here, we chose to take the latter approach because we wanted to explicitly model important elements of conformer generation, and this strategy has been successful. In our experience, modeling these specific geometric elements leads to a significant improvement in the quality of the output structures. It is still unclear if such an approach can tackle long-range interactions, which remains the major drawback of our model. We hope to answer this question in future work.