# OpenReview forum: "GeoMol: Torsional Geometric Generation of Molecular 3D Conformer Ensembles"
_NeurIPS.cc/2021/Conference — NeurIPS 2021 Spotlight_

### Official Review · Reviewer_DEuZ · 2021-07-01

**Rating:** 7
**Confidence:** 5

**Summary:**

This paper studies how to generate low-energy 3D molecular conformations based on 2D molecular graphs.
To this end, they propose the GEOMOL, an end-to-end trainable model that models molecular conformations in a SE(3)-invariant fashion, which first predicts the 3D local structures and then assembles them into final conformations with the predicted torsional angles.



**Limitations And Societal Impact:**

The main limitations of the work are two folds:

* (1) It only models the local structures (covalent bonds and their angles) and overlooks the long-range non-bonded angles, which may be crucial in multi-molecular systems or large molecules.

     Eq.4 (Line 273-274) indicates that GEOMOL predicts the torsional angle for each pair of LS independently. The benefit is that all LS can be assembled in parallel. However, it fails to take other structures into consideration (h_mol is used but it is not good enough).

* (2) The model relies on the torsional angles to assemble local structures, which means it fails to handle molecular systems with disconnected molecular graphs (e.g., molecular complexes).
One potential way is to combine GEOMOL with the previous distance-based models, i.e., first use GEOMOL to compute the structure of each connected component, and then use distance-based method (CGCF[1] and GraphDG[2]) or gradient-based (ConfGF[3], a related work on conformation generation, maybe it's better to cite it in paper) to assembly disconnected part.

**reference**
1. Xu, M., Luo, S., Bengio, Y., Peng, J., and Tang, J. Learning neural generative dynamics for molecular conformation generation.
2. Simm, G. and Hernandez-Lobato, J. M. A generative model for molecular distance geometry
3. Shi, Chence and Luo, Shitong and Xu, Minkai and Tang, Jian. Learning Gradient Fields for Molecular Conformation Generation

**Main Review:**

The GEOMOL takes torsional angles as well as stereochemistry (Tetrahedral chiral) into consideration and bypasses the 2-stage distance geometry technique used in previous works.

GEOMOL consists of several sub-modules, e.g., local Structure (1-hop) Prediction, Tetrahedral chiral corrections, Local Structure (LS) Assembly.
I find these sub-modules are overall novel at least in the ML community. The local structure assembly algorithm is the most interesting idea of the work in my view, which seems quite general and can be applied to other 3D modeling tasks.

The paper is generally well-written and the codes are provided.
Experimental results on GEOM datasets show that the performance of GEOMOL is quite promising.

**Questions**

 I wonder how well can each designed submodule perform on each task, especially on torsional angles prediction? (local structure prediction task and torsional angle prediction task).

**Update**
The authors address my concerns and looking forward to see the updated version of this exciting work.


**Time Spent Reviewing:**

2

---

> ### Author Response · Authors · 2021-08-10
> **Thank you**
>
> We thank the reviewer for the insightful comments and feedback. Here is our response:
>
> ---
> a. Regarding “how well each designed submodule can perform on each task, especially on torsional angles prediction?”:
> * Since we split our loss into five separate parts (see eq 8 from the appendix), we can directly check the performance of each submodule through these individual validation loss terms. We detail them here for the best models for the QM9/DRUGS datasets:
>
> 1-hop distance (MSE angstroms): 9.54e-4 / 3.10e-4
>
> 2-hop distance: (MSE angstroms): 3.90e-3 / 1.68e-3
>
> Bond angles: (cosine of angle differences): 1.00 / 1.00
>
>
> 3-hop: (MSE angstroms): 3.64e-2 / 3.10e-2
>
> Torsion angles: (cosine of angle differences): 0.85 / 0.85
>
> We can see that while the local model achieves a near-perfect prediction, the torsion model has room for improvement. We will focus future efforts on specifically improving this submodule of the model.
>
> ----
> b. Regarding the long-distance interactions:
> * Please see the detailed comments we gave to reviewer “c4sf”
>
> ----
> c. Regarding “molecular complexes” and use of ConfGF and other distance-geometry models:
> * We certainly agree with the proposed idea and look forward to combining ConfGF and GeoMol for predicting molecular complexes. Thank you for pointing it out.
>
> * We will certainly discuss and cite ConfGF and other recent models such as [1] in the updated version of our paper. GeoMol can be combined with some of the new techniques to lead to very powerful models.
>
> [1] An End-to-End Framework for Molecular Conformation Generation via Bilevel Programming, Xu et al, ICML 2021

---

### Official Review · Reviewer_2HaS · 2021-07-13

**Rating:** 7
**Confidence:** 5

**Summary:**

The authors propose the generative model that produces 3d conformations from a molecular graph based on predictions of local structures. First, for each anchor atom, the model predicts relative 1-hop positions of neighboring atoms, then predicts torsion angles. Finally, assemble the molecule in the deterministic way given predicted components. The authors comprehensively evaluate the model on the recently proposed dataset GEOM and compare it with state-of-the-art neural-based conformer generators and special software.

The paper's main contributions are: the authors propose a principal novel scheme for conformation generation that avoids expensive molecular geometry optimization used in current state-of-the-art neural conformer generators, still being translation, rotation, and reflection equivariant. The proposed model outperforms other baselines in both speeds of conformation generation and diversity of generated objects.


**Limitations And Societal Impact:**

The main limitation of the paper is the absence of generated conformations quality evaluations - how physically plausible the generated conformations in contrast with ground-truth conformations and sampled conformations from other baselines. The model could show the high diversity and coverage metrics, still producing conformations with physically implausible fragment structures. This evaluation can be easily done by computing the potential energies of conformations.

**Update**: The authors clarified this point in their rebuttal and provided an additional experiment demonstrating potential energies of generated structures.

**Main Review:**

**Originality:** The idea proposed in the paper is quite novel and differs from the current state-of-the-art neural methods for conformation generation both in used conformation representation and algorithm of generation: model operates with bond lengths, dihedral, and torsion angles instead of distance matrices and avoids time-consuming iterative process to optimize the final coordinates of conformation.

**Quality:** The authors carefully describe their motivation, goals and provide an overview of existing methods. However, the paper lacks a broad discussion of potential applications of the proposed model in the drug-discovery area.

**Clarity:** The paper is easy to follow. The equations and essential details of the generative process are clearly stated. Still, the paper reading experience suffers from the abundance of inline equations and the absence of equations references. Also, some subsections (i.e., 'Final LS assembly for a single bond' and 'How to compute γ?') are quite technical and can be moved fully or partially to the Appendix.

**Significance:** The experimental results show the gap in the coverage and diversity metrics between the proposed model and alternatives. Still, the metrics do not provide standard deviation for reported metrics, so it's hard to determine how significant this gap is.



**Time Spent Reviewing:**

6

---

> ### Author Response · Authors · 2021-08-10
> **We addressed all suggestions**
>
> We thank the reviewer for the insightful comments and feedback. We address all the suggestions below:
>
> ---
> 1) Regarding “potential energies of conformations to assess the physical plausibility of the generated conformations”:
>
> * We addressed this suggestion by computing potential energies as follows. For each conformer (without FF fine-tuning), we compute the energy with the MMFF force field from RDKit. Since we generate 2x the ground truth conformers per molecule, we save the x lowest energies and compute their mean/median. We then compute energies for all ground truth conformers and take their mean/median. Finally, we compute the difference between the mean/median of the generated conformers and of the ground truth conformers which gives a single number per molecule. We report below the overall median over all molecules of this aggregated relative energy:
>
> **QM9**
>
> GeoMol: **17.45/17.56**
>
> CGCF: 86.71/87.02
>
> GraphDG: 26.32/24.44
>
>
> **DRUGS**
>
> GeoMol: **83.72/77.52**
>
> CGCF: 1380.82/556.78
>
> GraphDG: 64350.58/61732.49
>
> The reader can notice that our model outperforms the ML models, sometimes by large margins, and especially for the larger molecules. We will make sure to include all the details and numbers in the final version of our paper.
>
> ----
> 2. Regarding “standard deviation for reported metrics”:
> * Confidence intervals are shown in appendix I, table 5. We will make sure to move them in the main text to improve visibility.
>
> ----
> 3. Regarding the broader discussion of potential applications of the proposed model in the drug-discovery area:
>
>   * We will enhance our existing discussion from lines 32-40 to detail the applications of GeoMol and how it can greatly assist the drug discovery (DD) pipeline.
>
>   * Specifically, we will describe how our method can be applied to the central DD problem of predicting drug - target interactions [1,2,3,4]. Concretely, GeoMol can be combined with an SE(3)-equivariant architecture (e.g. [7]) and used to explicitly model flexible body docking by jointly predicting both changes in the flexible conformer parts (i.e. usually the rotatable bonds and their torsion angles), as well as geometric and chemical 3D interactions (i.e. molecular docking poses). A property predictor based on 3D information (e.g. SchNet [5]) can additionally be utilized to evaluate or score the binding affinity of the resulting molecular complex candidates.
>
> * Moreover, GeoMol can be directly used to predict biological, chemical, and physical molecular properties since these are known to be a combination of all individual conformer properties [1]. Powerful property predictors can then be utilized for interpretable drug design [6].  In a preliminary set of experiments (to be published in a separate study), we have already seen the benefits of using GeoMol as a pre-training technique and intermediate representation for property predictors targeting biological activity.
>
> ----
> 4. Regarding readability of inline equations and technical subsections:
>
> * We thank the reviewer for pointing out these issues. We will surely incorporate the suggestions in the updated version of the paper. We hope to improve the reading experience of our paper.
>
>
> ----
> [1] Hawkins, P. C. D. Conformation generation: The state of the art., 2017
>
> [2] Schwab, C. H. Conformations and 3d pharmacophore searching., 2010
>
> [3] Predicting Drug–Target Interaction Using a Novel Graph Neural Network with 3D Structure-Embedded Graph Representation, Lim et al, 2019
>
> [4] Role of Molecular Dynamics and Related Methods in Drug Discovery, Vivo et al, 2015
>
> [5] SchNet: A continuous-filter convolutional neural network for modeling quantum interactions, Schutt et al, Advances in Neural Information Processing Systems, 2017
>
> [6] Multi-Objective Molecule Generation using Interpretable Substructures, Jin et al, ICML 2020
>
> [7] SE(3)-Transformers: 3D Roto-Translation Equivariant Attention Networks, Fuchs et al, 2020

---

> > ### Author Response · Authors · 2021-08-23
> > **re**
> >
> > Dear reviewer,
> >
> > We hope to have addressed all your comments and suggestions, but please let us know if there are other concerns.
> >
> > We wish that GeoMol's methodological advancements resulting in *significant quality improvements* compared to state-of-the-art ML models (both in terms of precision and recall), as well as the new experiments shown in this rebuttal might convince you to increase your rating.
> >
> >
> > Best regards,

---

> > > ### Author Response · Authors · 2021-08-26
> > > **reply**
> > >
> > > Dear reviewer,
> > >
> > > Thank you for reading our rebuttal.
> > >
> > > We would kindly like to ask if there are other questions or concerns that we could address in order to convince you to raise your current rating.
> > >
> > > Best regards,

---

### Official Review · Reviewer_c4sf · 2021-07-26

**Rating:** 7
**Confidence:** 4

**Summary:**

This paper introduces GeoMol, an end-to-end learning method for molecular conformation generation.  It is based on the construction and alignment of elements of local atomic structure, is efficient in its parameterization of geometric degrees of freedom, and is SE(3) invariant.  It achieves competitive performance on the GEOM-DRUGS and GEOM-QM9 tasks.

**Ethical Concerns:**

None.

**Limitations And Societal Impact:**

The authors convincingly address the limitations of the work.

There are no negative societal impacts to be addressed here.

**Main Review:**

Originality: To the best of my knowledge, the method is novel, and represents a fast and clean way of generating molecular conformations. In particular, the generation of elements of local structure and their subsequent assembly in an end-to-end manner allows for much more computationally efficient performance than previous methods of comparable performance.

Quality: The submission is technically sounds and the claims are backed up through experimental results.  The evaluation metrics are standard and well justified.

Clarity: The paper is very well written and self-contained.

Significance: I believe the results to be significant, especially as they allow for the rapid generation of many conformers with good accuracy.  One concern I have is whether this method will maintain accuracy for much larger molecules such as proteins, as these often are driven by long-distance interactions, and any local errors can compound.

*** Post-response update ***

Thank you for the additional information regarding your attempts to more explicitly model in long-distance interactions.  Seems like exciting future work!

**Time Spent Reviewing:**

4

---

> ### Author Response · Authors · 2021-08-10
> **Long range interactions**
>
> We thank the reviewer for the insightful feedback.
>
> Regarding the long-distance interactions:
>
> * We first note that our model does capture long range interactions via a global graph embedding for jointly predicting torsion angles (see eq 4), though we acknowledge here and in the paper that other more explicit mechanisms are desired for macromolecules.
>
> * We are exploring the following extensions of GeoMol that explicitly model long-range interactions, while keeping the end-to-end trainable property:
>    * i) We generalized our two-local-structure assembly procedure to work with k-hop paths by performing path local structure assembly, thus predicting k-hop distances in an end-to-end differentiable manner. However, this approach results in a noticeable slowdown in our training pipeline, and we are currently exploring various sampling strategies to avoid the expensive prediction of the full distance matrix.
>
>    * ii) To deal with the extra computational cost, we empirically proved that the k-hop distance of a path in 2D can be well approximated (with less than 5% error) by a transformer architecture that directly takes the bond distances and bond angles as inputs, a result that we wish to extend for 3D distances. This is a very promising avenue since the k-hop distance calculation is typically expensive to compute exactly.
>
>    * iii) We also recently extended the predictions of the $\alpha$ angles in eq 4 to use a transformer that jointly predicts all these angles using cross-attention.
>
>   * However, until the end time of this rebuttal, we could not obtain a full set of results for the above models, but we wish to keep the reviewers posted on this.

---

### Decision · Program_Chairs · 2021-09-27

**Decision:**

Accept (Spotlight)

**Comment:**

This paper introduces an end-to-end learning method called GeoMol for molecular conformation generation.  The algorithm is novel and provides  a fast and clean way of generating molecular conformations. It achieves competitive performance on the GEOM-DRUGS and GEOM-QM9 tasks. All the reviewers are excited about the work.